# VLMimic: Vision Language Models are Visual Imitation Learner for Fine-grained Actions

**Guangyan Chen**[1]    **Meiling Wang**[1]    **Te Cui**[1]    **Yao Mu**[2]    **Haoyang Lu**[1]

**Tianxing Zhou**[1]    **Zicai Peng**[1]    **Mengxiao Hu**[1]    **Haizhou Li**[1]    **Li Yuan**[3]    **Yi Yang**[1] [*]

**Yufeng Yue**[1] [*]

[1] **Beijing Institute of Technology**    [2] **The University of Hong Kong**    [3] **Peking University**

## Abstract

Visual imitation learning (VIL) provides an efficient and intuitive strategy for robotic systems to acquire novel skills. Recent advancements in Vision Language Models (VLMs) have demonstrated remarkable performance in vision and language reasoning capabilities for VIL tasks. Despite the progress, current VIL methods naively employ VLMs to learn high-level plans from human videos, relying on pre-defined motion primitives for executing physical interactions, which remains a major bottleneck. In this work, we present VLMimic, a novel paradigm that harnesses VLMs to directly learn even fine-grained action levels, only given a limited number of human videos. Specifically, VLMimic first grounds object-centric movements from human videos, and learns skills using hierarchical constraint representations, facilitating the derivation of skills with fine-grained action levels from limited human videos. These skills are refined and updated through an iterative comparison strategy, enabling efficient adaptation to unseen environments. Our extensive experiments exhibit that our VLMimic, using only 5 human videos, yields significant improvements of over 27% and 21% in RLBench and real-world manipulation tasks, and surpasses baselines by over 37% in long-horizon tasks. Code and videos are available at our home page.

## 1   Introduction

Visual Imitation Learning (VIL) has demonstrated remarkable efficacy in addressing various visual control tasks within intricate environments [1; 2; 3; 4; 5; 6; 7; 8; 9; 10]. Diverging from conventional approaches reliant on precise robot action labels, which often necessitates substantial human effort for data collection. Researchers increasingly turn to learning from human-object interaction videos that are easily accessible to reduce high data requirements.

Existing methods for skill acquisition leveraging video data can be broadly categorized into two classes. One typical approach learns efficient visual representations for robotic manipulation through self-supervised learning from large volumes of videos[11; 12; 13; 14; 15; 16; 17; 18; 19; 20**]. Another approach focuses on learning task-relevant priors to guide robot behaviors or derive a heuristic reward function for reinforcement learning [21; 14; 21; 22; 23; 24; 25; 26; 27; 28; 29].

---

[*]Yufeng Yue and Yi Yang are co-corresponding authors. This work was supported by the National Natural Science Foundation of China under Grant No. NSFC 62233002, 92370203. (email: yueyufeng@bit.edu.cn)

38th Conference on Neural Information Processing Systems (NeurIPS 2024).

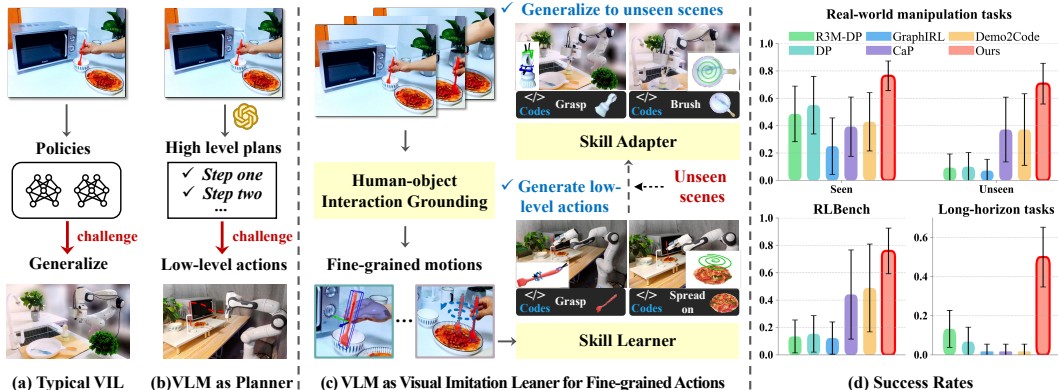

Figure 1: Illustration of our VLMimic. (a) Typical VIL methods struggle to generalize to unseen environments, and (b) current methods naively utilize VLMs as planners, encounter difficulties in generating low-level actions. (c) VLMimic grounds human videos to obtain action movements, and learns skills with fine-grained actions, while the skill adapter updates skills for generalization. (d) Our method achieves superior performance given a limited collection of human videos.

However, these approaches often encounter challenges when generalizing to unseen environments. Therefore, efficiently acquiring generalizable skills from limited videos remains highly challenging.

An appealing prospect for handling this challenge is to employ large pretrained models by encapsulating extensive prior knowledge from broad data. Recent advances in vision-language models (VLMs) provide particularly promising tools in this regard, with their emergent and fast-growing conceptual understanding, commonsense knowledge, and reasoning abilities. However, current VIL methods [30; 31; 32; 33; 34; 35; 36; 37; 38; 39; 40] naively employ VLMs to learn high-level plans, and typically rely on a repertoire of pre-defined motion primitives. This reliance on individual skill acquisition is often considered a major bottleneck of the system due to the lack of large-scale robotic data. The question then arises: *how can we leverage VLMs to learn even fine-grained action levels directly from human videos, eliminating the reliance on predefined primitives?*

However, adapting VLMs to achieve visual imitation learning for fine-grained actions is non-trivial due to the following critical reasons: (I) Lack of fine-grained action recognition ability. Despite existing advancements in VLMs, they still struggle to recognize low-level actions in videos. To overcome this obstacle, a human-object interaction grounding module is proposed, which parses videos into multiple segments, and estimates object-centric actions for subsequent analysis. Such that the intricate low-level action recognition task is converted into the pattern reasoning task, which is more tractable for existing VLMs. (II) Difficulty for VLMs in understanding motion signals. Motion signals are characterized by inherent redundancy, hindering models from extracting valuable information. To overcome this challenge, we propose hierarchical constraint representations for VLM reasoning, which exhibit semantic constraints through visualized actions and illustrate geometric constraints using keypoint values. This representation effectively reduces redundancy and facilitates a comprehensive understanding, enabling our method to learn skills from a limited set of human videos. (III) Disparities in demonstration and target scenes. Demonstration and execution scenes may involve different objects and tasks, impeding direct skill transfer. To this end, we propose a skill adapter with an iterative comparison strategy, which updates skills by iteratively contrasting with the demonstrated knowledge, facilitating the adaptation of learned skills to unseen scenes.

Based on the above analysis, we present VLMimic, an approach that employs VLMs to directly learn even fine-grained action levels from a limited number of human videos, and generalize to novel scenes. As shown in Fig. 1, our method parses videos into multiple segments and captures object-centric movements using the human-object interaction grounding module. Then, a skill learner employing hierarchical constraint representations extracts knowledge from estimated motions, deriving skills with fine-grained actions. In unseen environments, a skill adapter with an iterative comparison strategy revises and updates the learned skills based on observations and task instructions. Extensive experiments demonstrate that VLMimic achieves strong performance across various scenes, utilizing only 5 human videos without requiring additional training.

Our main contributions can be summarized as follows: (I) We propose VLMimic, a novel visual imitation learning framework empowered by VLMs, to learn generalizable robotic skills from

human demonstration videos. VLMimic features a skill learner for knowledge extraction and a skill adapter for iterative skill refinement, enabling efficient skill acquisition and adaptation. (II) We build an effective human-object interaction grounding algorithm to enhance fine-grained action recognition capabilities, and propose hierarchical constraint representations for VLM reasoning to reduce information redundancy and facilitate comprehensive action comprehension. (III) Our method outperforms other methods by over 27% on the RLBench. In real-world manipulation tasks, VLMimic achieves an improvement exceeding 21% in seen environments and 34% in unseen environments. Moreover, VLMimic exhibits an improvement of over 37% in long-horizon tasks.

## 2 Related Work

### 2.1 Learning from Human videos

Conventional learning approaches necessitate access to expert demonstrations, which include observations and precise actions for each timestep. Drawing on human capabilities, learning from observation offers efficient and intuitive methods for robots to develop new skills. A plethora of recent researches explore leveraging large-scale human video data to improve robot policy learning [11; 12; 13; 14; 15; 16; 17; 18; 19; 20; 41]. Representative methods, R3M [13] and MVP [12], which employ the internet-scale Ego4D dataset [11] to pretrain visual representations for subsequent imitation learning tasks. Another thread of work [21; 22; 23; 24; 25; 26; 27; 28; 29] focuses on learning task-relevant priors from videos to guide robot behaviors or derive a heuristic reward function for reinforcement learning. Learning by watching [27] learns human-to-robot translation, the resulting representations are used to guide robots to learn robotic skills. WHIRL [21] infers trajectories and interaction details to establish a prior, but it learns policy through real-world exploration and requires a large number of rollouts to converge. GraphIRL [24] performs graph abstraction on the videos followed by temporal matching to measure the task progress, and a dense reward function is employed to train reinforcement learning algorithms. Despite these advancements, acquiring generalizable skills efficiently from limited demonstration videos remains highly challenging.

### 2.2 Visual Imitation Learning with VLMs

Motivated by the notable success of VLMs across various domains, recent research [32; 33; 34; 35; 36; 37] investigate their potential in VIL. GPT-4V for Robotics [33] analyzes videos of humans performing tasks and outputs robot programs that incorporate insights into affordances. Digknow [32] distills generalizable knowledge with a hierarchical structure, enabling the effective generalization to novel scenes. Demo2code [37] generates robot task code from demonstrations via an extended chain-of-thought and defines a common latent specification to connect the two. VLaMP [34] predicts visual planning from videos through video action segmentation and forecasting, handling long video history and complex action dependencies. However, these approaches often rely on predefined movement primitives or pre-trained skills to execute lower-level actions, thereby only partially solving the control stack. In contrast, our investigation aims to push these boundaries and learn all lower-level actions for the robot, eliminating the reliance on predefined primitives.

## 3 VLMimic

Considering video demonstrations $\mathcal{V}$ of a human performing manipulation tasks, recorded using an RGB-D camera. The overall pipeline of VLMimic is illustrated in Fig. 2. Our method first grounds human videos, segmenting them into subtask intervals $\{\tau_i\}_{i=1}^{V}$ and capturing object-centric interactions $I$. A skill learner with hierarchical representations then extracts knowledge from the obtained interactions, deriving skills with fine-grained actions. In unseen environments, a skill adapter employs an iterative comparison strategy to revise and update the learned skills based on observations and task instructions.

### 3.1 Human-object Interaction Grounding

Despite VLMs demonstrating proficiency in various vision tasks, they still struggle with fine-grained action recognition within videos. To mitigate this limitation, a four-stage process, illustrated in Fig. 3,

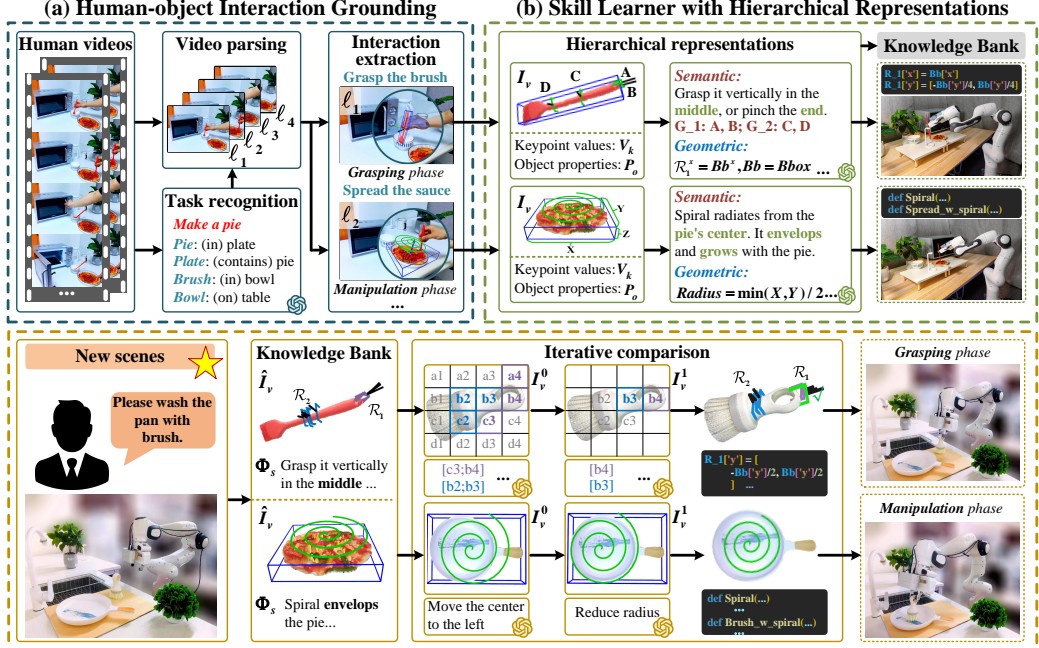

**(c) Skill Adapter with Iterative Comparison**

Figure 2: Illustration of our VLMimic. (a) The human-object interaction grounding module parses videos into multiple segments and captures object-centric movements. Then, (b) a skill learner extracts knowledge from action motions and derives skills. In novel scenes, (c) a skill adapter updates the learned skills to facilitate adaptation.

is utilized to extract object-centric interactions for skill learning, transforming this intricate problem into pattern reasoning problems, typically more tractable for existing VLMs.

**Task recognition**. Keyframes $\mathcal{K}$ are intermittently extracted from videos $\mathcal{V}$, vision foundation models VFM [42; 43; 44] are utilized to detect objects within these frames. Utilizing keyframes $\mathcal{K}$ and textual detection results $\boldsymbol{T}_d$, VLMs are instructed to transcribe videos into task instructions $\boldsymbol{T}_t$, and compile the task-related objects $\boldsymbol{T}_o$ into textual information. The object information is predicated on the initial frame of the video data, comprising a list of object names and their spatial relationships. The task recognition procedure is formulated as follows:

$$\boldsymbol{T}_d = \text{VFM}(\mathcal{K}), \quad \boldsymbol{T}_t, \boldsymbol{T}_o = \text{VLM}(\boldsymbol{T}_d, \mathcal{K}). \tag{1}$$

**Video parsing**. Videos are parsed into segments $\{\boldsymbol{\tau}_i\}_{i=1}^V$, using interaction markers that identify interaction periods. SAM-Track [45; 46; 47; 48; 49] predicts hand and task-related object masks for each frame, and corresponding point clouds $\mathcal{P}$ are generated through back-projection. Markers are then identified by determining the interaction start time $\boldsymbol{t}_i$ and end time $\boldsymbol{t}_e$, partitioning videos $\mathcal{V}$ into multiple segments. Segments with hand motion trajectory lengths below than $\gamma$ are filtered out, yielding final set of segments $\{\boldsymbol{\tau}_i\}_{i=1}^V$. Concretely, the interaction markers are obtained as follows:

$$\boldsymbol{d} = \text{dist}(\mathcal{P}), \quad \boldsymbol{t}_i = \{t | \boldsymbol{d}^{t-1} > \epsilon \wedge \boldsymbol{d}^t < \epsilon\}, \quad \boldsymbol{t}_e = \{t | \boldsymbol{d}^{t-1} < \epsilon \wedge \boldsymbol{d}^t > \epsilon\}, \tag{2}$$

where function dist calculates the distance between any two point clouds.

**Subtask recognition**. Each segment $\boldsymbol{\tau}_i$ is analyzed by VLMs, which generate a subtask textual description $\boldsymbol{T}_{\tau_i}$, and categorize the segment into grasping or manipulation phases based on the interacting entities and $\boldsymbol{T}_{\tau_i}$. VLMs also identify master objects $\boldsymbol{O}_m$ and slave objects $\boldsymbol{O}_s$. In the grasping phase, the agent performs a reach-and-grasp action targeting $\boldsymbol{O}_m$, designating the hand as $\boldsymbol{O}_s$. In the manipulation phase, the agent employs $\boldsymbol{O}_s$ to interact with $\boldsymbol{O}_m$.

**Object-centric interaction extraction**. FrankMocap [50] and the Iterative Closest Point (ICP) algorithm [51; 52] are employed to derive precise hand pose trajectories, which are subsequently converted into robot gripper pose trajectories. Furthermore, BundleSDF [53] is employed for object reconstruction, and FoundationPose [54] is leveraged for object pose estimation based on reconstructed objects $\boldsymbol{O}$. In grasping phases, interactions $\boldsymbol{I}$ are represented as grasp poses at hand-object contacts. For manipulation phases, $\boldsymbol{I}$ are defined as trajectories of slave objects $\boldsymbol{O}_s$ relative to

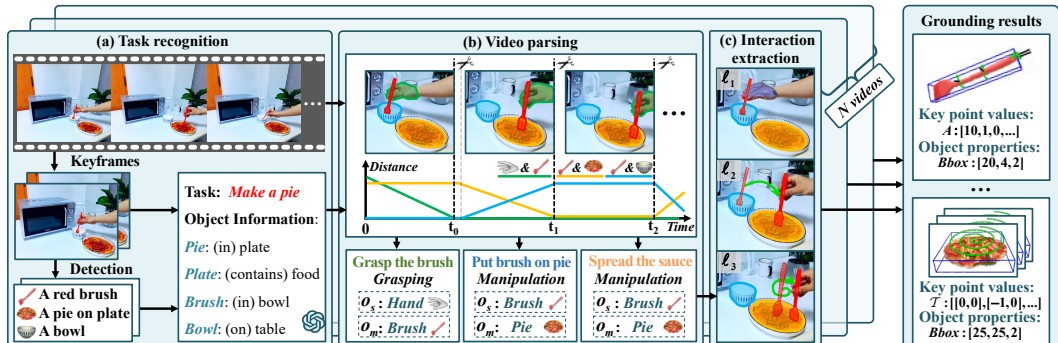

Figure 3: Illustration of Human-object interaction grounding module. (a) It recognizes tasks and related objects from human videos, (b) parses videos into multiple segments based on this information, and subsequently (c) identifies object-centric interactions within each segment.

master objects $O_m$. This object-centric paradigm facilitates efficient skill acquisition and enables VLMimic to accommodate demonstrations across diverse viewpoints.

## 3.2 Skill Learner with Hierarchical Representations

A straightforward approach for learning skills involves directly discerning the numerical trajectory patterns [55; 56]. However, VLMs face challenges in reasoning about inherently redundant motion signals, limiting their ability to extract valuable information. To reduce redundancy and foster comprehensive comprehension, hierarchical constraint representations are proposed for skill learning, as illustrated in Fig. 2. These representations exhibit semantic constraints via visualized interaction $I_v$ and further detail the fine-grained geometric constraints by integrating keypoint values $V_k$.

**Learning with hierarchical constraint representations**. Rendering interaction $I$, and textual notations $T_n$ on objects $O$ to derive visualized interaction $I_v$, VLMimic facilitates reasoning capabilities to analyze semantic constraints $\Phi_s$ by encouraging VLMs to attend to objects and their related actions, and integrating keypoint values $V_k$ and object properties $P_o$ (e.g., 3-D bounding boxes) to derive geometric constraints $\Phi_g$. Formally, constraints are learned as follows:

$$I_v = \text{Render}([I, T_n], O), \quad \Phi_s = \text{S}_1(I_v) \quad \Phi_g = \text{G}_1(\Phi_s, V_k, P_o), \tag{3}$$

where $\text{S}_1$, and $\text{G}_1$ are functions to learn semantic, and geometric constraints, respectively.

(I) Grasping constraints. Inspired by task space regions (TSRs) [57], the grasping constraint $\Phi_g$ can be approximated as a series of bounded regions $\{\mathcal{R}_i\}_{i=1}^{N_C}$. Interactive grasp poses $I$ are exhibited on objects, each associated with an index notation $T_n$. These visualized interactions $I_v$ are presented to VLMs, leveraging their inherent knowledge and visual understanding ability to summarize semantic constraints $\Phi_s$ and group these poses. Geometric constraints $\Phi_g$, represented as bounded regions, are derived by calculating ranges of grasp pose values $V_k$ within the same group, and associating them with object properties $P_o$. This approach simplifies the complex task of constraint region generation into a series of visual understanding based multiple-choice question answering. Moreover, representing constraints through object properties enhances generalization across objects.

(II) Manipulation constraints. Interaction trajectory $I$ is delineated on the master object $O_m$, incorporating keypoints $V_k$ in the textual prompt. Semantic constraints $\Phi_s$ are identified by VLMs based on the visualized interaction $I_v$ and subtask description $T_{\tau_i}$. Geometric constraints $\Phi_g$ are then formulated based on semantic constraints $\Phi_s$, keypoint values $V_k$, and object properties $P_o$, expressing $\Phi_g$ via the trajectory code. The code comprises two components: parameter estimation functions $f_p$, which derives trajectory parameters from object properties, and trajectory generation functions $f_s$, employing estimated parameters to generate a sequence of slave object poses relative to the master object, promoting effective generalization across various objects and spatial configurations.

During execution, grasp candidates are uniformly sampled within the learned grasping constraints, and object-centric trajectories predicted from manipulation constraints are converted to end-effector trajectories in the world frame using each grasp candidate of the slave object and object poses of master and slave objects. The resulting end-effector trajectory candidates are evaluated using motion planner, such as OMPL [58], the trajectory with the highest fraction is selected.

**Knowledge bank construction**. A knowledge bank $\boldsymbol{B}$ is established to archive both high-level planning and low-level skill insights, storing knowledge with key-value pairs $(\boldsymbol{k}_i, \boldsymbol{v}_i)$. High-level planning knowledge is indexed using task description $\boldsymbol{T}_t$ as keys, paired with the consequent action sequence $\boldsymbol{T}_\tau$ as values. For low-level skill knowledge, keys are constituted by the object images and subtask description $\boldsymbol{T}_{\tau_i}$, and values comprise reconstructed objects, as well as semantic constraints $\boldsymbol{\Phi}_s$ and geometric constraints $\boldsymbol{\Phi}_g$ representing learned skills.

## 3.3 Skill Adapter with Iterative Comparison

Even though the skill learner exhibits efficient skill acquisition, the demonstration and execution scenes may differ in objects and tasks, impeding direct skill transfer to unseen environments. To mitigate these challenges, VLMs are instructed to adapt skills via an iterative comparison strategy, as depicted in Fig. 2. This approach updates learned skills by iteratively contrasting with the demonstrated knowledge, thereby enabling effective adaptation of retrieved skills to novel scenes.

**High level planning**. High-level planning knowledge $\boldsymbol{T}_\tau$ is retrieved from knowledge bank $\boldsymbol{B}$ based on the task instruction, which acts as the in-context example for VLMs, along with the scene observation. VLMs serve as a physically-grounded task planner [59; 60], generating a sequence of actionable steps and descriptions of task-related objects $\boldsymbol{T}_o$.

**Iterative comparison**. In each iteration, VLMs perform a comparative analysis between the adapted interaction $\boldsymbol{I}$ and retrieved interaction $\hat{\boldsymbol{I}}$, subsequently updating the skill constraints $\boldsymbol{\Phi}_s$ and $\boldsymbol{\Phi}_g$. This iterative process persists until either convergence is achieved or the maximum number of iterations $N_I$ is reached. This approach facilitates reasoning in VLMs by directing their attention to discrepancies, and enables VLMs to pinpoint the best available solution through an iterative process. The adapting procedure at the $i$-th iteration can be formally represented as:

$$\boldsymbol{I}_v^i = \text{Render}(\boldsymbol{\Phi}_g^i, O), \quad \boldsymbol{\Phi}_s^{i+1} = \text{S}_\text{a}(\hat{\boldsymbol{I}}_v, \boldsymbol{I}_v^i, \hat{\boldsymbol{\Phi}}_s, \boldsymbol{\Phi}_s^i), \quad \boldsymbol{\Phi}_g^{i+1} = \text{G}_\text{a}(\hat{\boldsymbol{\Phi}}_g, \boldsymbol{\Phi}_g^i, \boldsymbol{\Phi}_s^{i+1}, \boldsymbol{V}_k, \boldsymbol{P}_o), \quad (4)$$

where $\hat{\boldsymbol{\Phi}}_g$ and $\hat{\boldsymbol{\Phi}}_s$ denote referential constraints, extracted from the knowledge base. The functions $\text{S}_\text{a}$ and $\text{G}_\text{a}$ adapt semantic and geometric constraints, respectively.

(I) Grasping constraint adaptation. As the grasping orientation is typically derivable from position constraints using grasping models [61; 62; 63], our work focuses on transferring position constraints. The visualized grasping position space is discretized into an $m \times n$ grid ($m, n \in Z$) and annotated with textual notations $\boldsymbol{T}_n$, obtaining $\boldsymbol{I}_v^0$. VLMs are instructed to update semantic constraints $\boldsymbol{\Phi}_s$, by contrasting with the referential interaction $\hat{\boldsymbol{I}}_v$ and semantic constraints $\hat{\boldsymbol{\Phi}}_s$, and to adapt geometric constraints $\boldsymbol{\Phi}_g$ by sampling $K$ outputs of grasping region selection. The updated $\boldsymbol{\Phi}_g$ are then visualized for the next iteration. The 3-D positional region is represented using two perspectives, and the consistency of the selected regions for the overlapping area validates the VLM outputs. The obtained constraints are expressed via object properties to enhance generalization.

(II) Manipulation constraint adaptation. VLMs are instructed to iteratively self-summarise and update manipulation constraints based on the task instruction and scene differences. VLMimic generates trajectories adhering to geometric constraints $\boldsymbol{\Phi}_g$, which are exhibited on master objects. VLMs are instructed to analyze the deviation of the adapted interaction $\boldsymbol{I}_v$ from the referential interaction $\hat{\boldsymbol{I}}_v$ to revise semantic constraints $\boldsymbol{\Phi}_s$, and geometric constraints $\boldsymbol{\Phi}_g$ undergo refinement predicated on the updated $\boldsymbol{\Phi}_s$, along with trajectory keypoint values $\boldsymbol{V}_k$ and object properties $\boldsymbol{P}_o$.

**Failure reasoning**. Despite the ability of VLMs to generate effective constraints, environmental noise, such as trajectory estimation errors, impedes successful task execution. Thus, we leverage VLMs to detect and address failures during execution by providing them with perceptual results, such as object pose and robot end-effector trajectories, enabling autonomous failure identification and reasoning for rectification.

# 4 Experiments

**Baselines**. VLMimic is compared with five representative methods: (1) R3M-DP that utilizes the pre-trained R3M visual representation [13] with the state-of-the-art (SOTA) diffusion policy [7]; (2) Diffusion Policy (DP) [7], a SOTA end-to-end policy method; (3) GraphIRL [24], a method that employs graph abstraction and learns reward functions for reinforcement learning (RL); (4) Code

Table 1: Success rates on RLbench. "Obs-act", "Template", and "Video" indicate paired observation-action sequences, code templates, and videos performing subtasks.

| Methods | R3M-DP | DP | GraphIRL | CaP | Demo2Code | Ours |
|---|---|---|---|---|---|---|
| Overall | 0.13($\pm$0.12) | 0.15($\pm$0.13) | 0.12($\pm$0.12) | 0.44($\pm$0.33) | 0.49($\pm$0.32) | **0.76($\pm$0.17)** |

| Methods | Type of demos | Num of demos | Reach target | Take lid off saucepan | Pick up cup | Toilet seat up | Open box | Open door |
|---|---|---|---|---|---|---|---|---|
| R3M-DP | Obs-act | 100 | 0.37 | 0.20 | 0.20 | 0.07 | 0.02 | 0.25 |
| DP | Obs-act | 100 | 0.43 | 0.25 | 0.24 | 0.05 | 0.04 | 0.22 |
| GraphIRL | Video | 100 | 0.39 | 0.14 | 0.23 | 0.03 | 0.03 | 0.21 |
| CaP | Template | 5 | 0.95 | 0.90 | 0.58 | 0.05 | 0.12 | 0.65 |
| Demo2Code | Video | 5 | 0.94 | 0.86 | 0.65 | 0.06 | 0.19 | 0.83 |
| **Ours** | **Video** | **5** | **0.97** | **0.94** | **0.80** | **0.76** | **0.75** | **0.90** |

| Methods | Type of demos | Num of demos | Meat off grill | Open drawer | Open grill | Open microwave | Open oven | Knife on board |
|---|---|---|---|---|---|---|---|---|
| R3M-DP | Obs-act | 100 | 0.15 | 0.25 | 0.07 | 0.03 | 0.00 | 0.00 |
| DP | Obs-act | 100 | 0.17 | 0.28 | 0.09 | 0.07 | 0.00 | 0.00 |
| GraphIRL | Video | 100 | 0.16 | 0.18 | 0.04 | 0.04 | 0.02 | 0.00 |
| CaP | Template | 5 | 0.35 | 0.17 | 0.46 | 0.12 | 0.16 | 0.78 |
| Demo2Code | Video | 5 | 0.57 | 0.22 | 0.40 | 0.14 | 0.21 | 0.79 |
| **Ours** | **Video** | **5** | **0.79** | **0.75** | **0.81** | **0.45** | **0.43** | **0.76** |

as Policy (CaP) [64], an LLM-driven method that re-composes API calls to generate new policy code; and (5) Demo2code [37], an LLM-driven planner method that translates demonstrations into task code. We modify it to integrate the analysis results from GPT-4V for Robotics [33], enabling it to transcribe videos into code. R3M-DP and DP are trained using the robot demonstrations with paired observation and action sequences. GraphIRL is trained in simulators with paired robot videos, Demo2code and our method learns skills with human videos in real-world experiments and robot videos in simulation experiments.

## 4.1 Simulation Manipulation Tasks

**Experimental setup**. To assess our approach on challenging robotic manipulation tasks, the RLBench [65] benchmark is utilized for simulation tasks. Due to the unavailability of human videos in simulations, demo2code and our method utilize robot videos captured from a single-camera perspective during demonstrations, incorporating robot gripper trajectories.

**Results**. We investigate the capacity of VLMimic to acquire skills from a limited collection of video demonstrations, without requiring additional training. Our evaluation encompasses 12 manipulation tasks, as detailed in Table 1, demonstrating that our method surpasses all other methods in 11 out of these tasks. Our method, learned with only 5 human videos, obviously outperforms R3M-DP and DP by over 61% in overall performance, despite both being trained on 100 robot demonstrations. Compared to CaP and demo2code, our method demonstrates an improvement exceeding 27%, highlighting the significant performance enhancements facilitated by the VLMimic framework.

## 4.2 Real-world Manipulation Tasks

**Experimental setup**. The real-world testing environment (E) is divided into "seen" (SE) and "unseen" (UE) categories. The "seen" category allows for testing in the environment where demonstrations were collected, whereas the "unseen" category involves testing in a distinct environment characterized by different objects and layouts. Success criteria are human-evaluated and the success rate is calculated from 10 randomized object positions and orientations.

**Results**: To validate the effectiveness of VLMimic in real-world settings, we conduct experiments involving 14 challenging real-world manipulation tasks selected from recent robotics research [66; 67; 4; 68]. Quantitative results, presented in Table 2, demonstrate that VLMimic clearly outperforms other methods across all tasks, particularly in the "unseen" environment (UE). VLMimic achieves an

Table 2: Success rates on real-world manipulation experiments. "Obs-act", "Template", and "Video" indicate paired observation-action sequences, code templates, and videos performing subtasks. "SE" and "UE" are seen and unseen environments.

| Methods | R3M-DP | DP | GraphIRL | CaP | Demo2Code | Ours |
|---|---|---|---|---|---|---|
| Overall (SE) | 0.49(±0.20) | 0.55(±0.21) | 0.25(±0.21) | 0.39(±0.22) | 0.43(±0.21) | **0.76(±0.11)** |
| Overall (UE) | 0.09(±0.10) | 0.10(±0.10) | 0.07(±0.08) | 0.37(±0.24) | 0.37(±0.26) | **0.71(±0.15)** |

| Methods | Type of demos | Num of demos | Open drawer | | Stack block | | Open oven | | Put fruit on plate | | Press button | | Open microwave | | Put tray in oven | |
|---|---|---|---|---|---|---|---|---|---|---|---|---|---|---|---|---|
| | | | SE | UE | SE | UE | SE | UE | SE | UE | SE | UE | SE | UE | SE | UE |
| R3M-DP | Obs-act | 100 | 0.2 | 0.1 | 0.6 | 0.2 | 0.3 | 0.0 | 0.8 | 0.3 | 0.7 | 0.2 | 0.2 | 0.0 | 0.4 | 0.0 |
| DP | Obs-act | 100 | 0.3 | 0.1 | 0.6 | 0.2 | 0.4 | 0.1 | 0.9 | 0.4 | 0.7 | 0.1 | 0.3 | 0.0 | 0.4 | 0.0 |
| GraphIRL | Video | 100 | 0.2 | 0.0 | 0.4 | 0.1 | 0.0 | 0.0 | 0.7 | 0.2 | 0.4 | 0.2 | 0.0 | 0.0 | 0.2 | 0.0 |
| CaP | Template | 5 | 0.3 | 0.3 | 0.5 | 0.5 | 0.3 | 0.2 | 0.8 | 0.8 | 0.7 | 0.7 | 0.1 | 0.1 | 0.2 | 0.1 |
| Demo2Code | Video | 5 | 0.3 | 0.3 | 0.5 | 0.4 | 0.3 | 0.3 | 0.8 | 0.9 | 0.8 | 0.8 | 0.2 | 0.1 | 0.3 | 0.2 |
| Ours | Video | 5 | **0.8** | **0.7** | **0.9** | **0.8** | **0.6** | **0.6** | **0.9** | **0.9** | **0.8** | **0.9** | **0.7** | **0.6** | **0.7** | **0.7** |

| Methods | Type of demos | Num of demos | Turn on oven | | Sweep table | | Insert box | | Brush pan | | Sauce spread | | Put toy to drawer | | Pour from cup to cup | |
|---|---|---|---|---|---|---|---|---|---|---|---|---|---|---|---|---|
| | | | SE | UE | SE | UE | SE | UE | SE | UE | SE | UE | SE | UE | SE | UE |
| R3M-DP | Obs-act | 100 | 0.2 | 0.0 | 0.7 | 0.2 | 0.4 | 0.0 | 0.6 | 0.1 | 0.6 | 0.1 | 0.6 | 0.1 | 0.5 | 0.0 |
| DP | Obs-act | 100 | 0.3 | 0.0 | 0.8 | 0.1 | 0.3 | 0.1 | 0.7 | 0.1 | 0.7 | 0.0 | 0.7 | 0.1 | 0.6 | 0.1 |
| GraphIRL | Video | 100 | 0.2 | 0.1 | 0.5 | 0.2 | 0.0 | 0.0 | 0.2 | 0.0 | 0.2 | 0.1 | 0.4 | 0.1 | 0.1 | 0.0 |
| CaP | Template | 5 | 0.3 | 0.3 | 0.6 | 0.5 | 0.1 | 0.1 | 0.3 | 0.4 | 0.3 | 0.3 | 0.6 | 0.7 | 0.4 | 0.2 |
| Demo2Code | Video | 5 | 0.2 | 0.1 | 0.6 | 0.6 | 0.3 | 0.2 | 0.4 | 0.3 | 0.3 | 0.4 | 0.7 | 0.6 | 0.3 | 0.2 |
| Ours | Video | 5 | **0.8** | **0.7** | **0.9** | **0.9** | **0.6** | **0.4** | **0.8** | **0.7** | **0.8** | **0.7** | **0.8** | **0.8** | **0.6** | **0.5** |

Table 3: Success rates on long-horizon tasks. "Obs-act", "Template", and "Video" indicate observation-action sequences, code templates, and videos performing tasks.

| Methods | Type of demos | Num of demos | Make coffee | Clean table | Make a pie | Wash pan | Make slices | Chem. exp. | Overall |
|---|---|---|---|---|---|---|---|---|---|
| R3M-DP | Obs-act | 100 | 0.10 | 0.30 | 0.20 | 0.10 | 0.00 | 0.10 | 0.13(±0.09) |
| DP | Obs-act | 100 | 0.00 | 0.20 | 0.10 | 0.00 | 0.10 | 0.00 | 0.07(±0.07) |
| GraphIRL | Video | 100 | 0.00 | 0.10 | 0.00 | 0.00 | 0.00 | 0.00 | 0.02(±0.04) |
| CaP | Template | 5 | 0.00 | 0.10 | 0.00 | 0.00 | 0.00 | 0.00 | 0.02(±0.04) |
| Demo2Code | Video | 5 | 0.00 | 0.10 | 0.00 | 0.00 | 0.00 | 0.00 | 0.02(±0.04) |
| **Ours** | **Video** | **5** | **0.40** | **0.70** | **0.70** | **0.40** | **0.50** | **0.30** | **0.50(±0.15)** |

improvement exceeding 21% in SE and more than 34% in UE. Results reveal the outstanding ability of VLMimic to acquire skills from human videos and adapt them to unseen environments.

## 4.3 Real-world Long-Horizon Tasks

**Experimental setup**. Since baseline methods struggle to complete long-horizon tasks in the UE setting, experiments are conducted in the SE setting. All other experimental settings are consistent with those in the real-world manipulation task.

**Results**. The performance of VLMimic on long-horizon tasks is quantitatively evaluated by its successful completion of six distinct tasks, each comprising at least five subtasks. Experimental results, as depicted in Table 3, obviously exhibit a substantial enhancement achieved by our method over baseline methods. These outcomes suggest that the proposed method is capable of developing robust skills, thereby achieving promising performance in even long-horizon tasks.

## 4.4 Robustness against viewpoint variance

The keypoint-centric representation approach enables our method to tolerate different observational perspectives. To demonstrate the robustness of our method to varying viewpoints. Experiments are conducted in real-world unseen environments, utilizing distinct viewpoints, as shown in Figure 4, where the first angle serves as the default perspective used in our experiments. Experimental results shown in Table 4 prove that our method exhibits only a 7% fluctuation in performance under varying viewpoints, demonstrating the resilience of VLMimic to viewpoint changes.

Table 4: Robustness against viewpoint variance.

| Methods | Viewpoint 1 | Viewpoint 2 | Viewpoint 3 | Viewpoint 4 |
|---------|-------------|-------------|-------------|-------------|
| Ours | 0.71($\pm$0.15) | 0.67($\pm$0.16) | 0.70($\pm$0.15) | 0.64($\pm$0.17) |

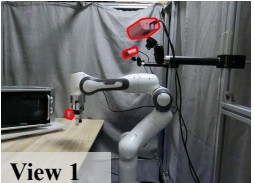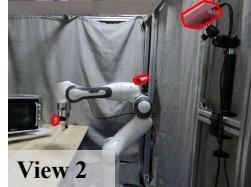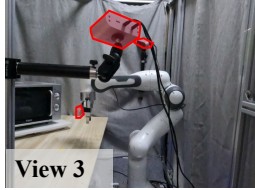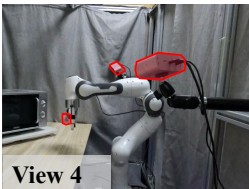

Figure 4: Configuration of various viewpoints.

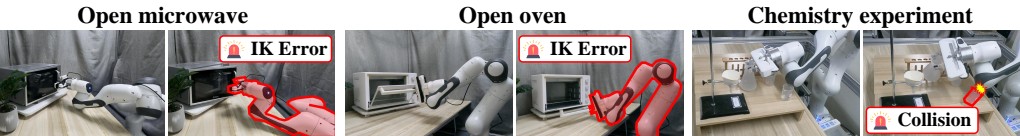

Figure 5: Examples of failure cases.

## 4.5 Real-world failure cases

Figure 5 elucidates scenarios that present significant challenges for resolution through VLM reasoning. These scenarios encompass: (I) The task execution may exceed the hardware limitations of the physical robot, inducing inverse kinematics (IK) errors. (II) Incomplete environmental perception increases the risk of obstacle collisions, leading to task failure. Since the training datasets for VLMs exhibit a significant lack of data related to robot dynamics, these models lack associated knowledge, exhibiting a limited capacity for error analysis and struggling to infer correction strategies when confronted with these failures.

## 4.6 Ablation Studies

Comprehensive ablation studies are conducted to investigate the fundamental designs of our VLMimic approach. The effects of these design decisions are assessed by measuring the success rate on real-world manipulation tasks, which is computed across 10 randomized object positions and orientations.

**Hierarchical constraint representations**. Table 5 (a) analyzes three distinct constraint representations. Variants that exclusively reason semantic constraints or directly obtain geometric constraints without semantic analysis, lead to diminished performance. The results exhibit that hierarchical constraint representations enhance skill acquisition capabilities, demonstrating the pivotal role in facilitating the understanding and reasoning capabilities of VLMs.

**Grasping learning**. Table 5 (b) presents variants and their respective performance. The first variant utilizes VLMs for direct prediction of constraint region values, resulting in a significant performance decline. The second variant employs the DBScan clustering algorithm to group grasp poses and derive constraints as bounded regions. However, this method only considers numerical distributions without incorporating grasping common sense, leading to performance degradation.

**Number of human videos**. Table 5(c) presents an analysis of the impact of human video quantity on performance. Results indicate that our method attains high success rates on complex tasks with a single human video demonstration, and increasing the number of videos yields performance gains. The results show that our approach can efficiently learn generalizable skills from a limited number of human videos. We choose to use 5 demonstration videos to balance data availability and performance.

**Comparison strategy**. Table 5 (d) analyzes the impact of the comparison strategy in skill adapters. Variants compare constraints exclusively utilizing either visualized interactions or keypoints exhibit decreased success rates. Visual comparison facilitates semantic contrast in VLM, while keypoint values provide fine-grained geometric information. Experimental results illustrate that our strategy facilitates reasoning for both semantic and geometric constraint adaptation.

**Number of iterations**. We conduct an analysis on the impact of iteration count in skill adapter and search for the optimal choice, as shown in Table 5(e). Reducing the number of iterations to 0 results in

Table 5: Ablation experiments with VLMimic on real-world manipulation experiments. "SE" and "UE" are seen and unseen environments. Default settings are marked in gray.

(a) Hierarchical representations.

| Variants | SE |
| --- | --- |
| Geometric constraints | 0.61 |
| Semantic constraints | 0.68 |
| Hierarchical constraints | 0.76 |

(b) Grasping learning.

| Variants | SE |
| --- | --- |
| Value prediction | 0.52 |
| Grouping (DBSCAN) | 0.59 |
| Grouping with VLMs | 0.76 |

(c) Number of videos.

| Number | SE | Number | SE |
| --- | --- | --- | --- |
| 1 | 0.68 | 7 | 0.67 |
| 3 | 0.72 | 9 | 0.78 |
| 5 | 0.76 | 11 | 0.78 |

(d) Comparison strategy.

| Variants | UE |
| --- | --- |
| Visual comparison | 0.61 |
| Keypoint comparison | 0.60 |
| Visual with keypoints | 0.71 |

(e) Number of iterations.

| Number | UE | Number | UE |
| --- | --- | --- | --- |
| 0 | 0.58 | 3 | 0.68 |
| 1 | 0.61 | 4 | 0.71 |
| 2 | 0.66 | 5 | 0.71 |

(f) Failure reasoning.

| Number | UE | Number | UE |
| --- | --- | --- | --- |
| 0 | 0.65 | 3 | 0.72 |
| 1 | 0.68 | 4 | 0.70 |
| 2 | 0.71 | 5 | 0.71 |

a noticeable decrease in performance. Strong results are observed in the initial iteration, with modest improvements in subsequent iterations. The findings indicate that this iterative approach enhances the effectiveness of skill adaptation by enabling VLMs to identify the best available solution. For enhanced performance, 4 iterations are selected.

**Failure reasoning**. The impact of failure reasoning is investigated in Table 5(f). The success rate exhibits an upward trend with increasing iterations, reaching an elbow point at 2 iterations, providing an optimal trade-off between real-time performance and success rate. Failure reasoning proves crucial for tasks demanding high-precision manipulation, which are susceptible to environmental noise. It enhances both the success rate and the robot's ability to operate in intricate environments.

## 5 Conclusion

In this paper, we present VLMimic, a novel approach that harnesses VLMs to learn skills with even fine-grained action levels from a limited number of human videos, and effectively generalize them to unseen environments. VLMimic first extracts object-centric interactions from human videos, and learns skills based on these interactions, using hierarchical constraint representations. In unseen environments, these skills are updated through an iterative comparison strategy. Extensive experiments conducted on various manipulation and challenging long-horizon tasks demonstrate the superior performance achieved by our VLMimic, utilizing only 5 human videos without requiring additional training, exhibiting strong skill acquisition and adaptation capabilities.

**Limitations**. Despite the promising performance exhibited by VLMimic, current VLMs are still limited by inference latency and computational resource requirements. We believe that the progression of lightweight VLMs will mitigate these limitations.

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

# A  Implementation details

In human-object interaction grounding module, the Tokenize Anything [44] model is employed during task recognition to improve fine-grained scene understanding ability. The textual detection results are integrated using VLMs to generate concise task descriptions and detailed object information. The videos are segmented using a threshold $\epsilon$ of $2cm$. Segments with hand motion trajectory lengths below $\gamma = 10cm$ are discarded. During the grasping constraint learning phase, the number of regions $N_c$ is automatically determined by the VLMs. In manipulation constraint learning, keypoints are obtained by uniformly sampling 10 points. For the skill adapter, the maximum number of iterations is set to $N_I = 4$. During grasping constraint adaptation, visualized grasping position space is discretized into a $5 \times 5$ grid, with $K = 4$ outputs sampled per iteration.

During skill execution, the pretrained Grounded-segment-any-parts model [69; 70] is used to generate segmentation maps of queried objects or parts. These segmentation maps are then utilized to predict object-centric pose sequences using codes generated by VLMs. FoundationPose [54] is employed to track object poses, transforming the object-centric poses into the world frame. The robotic arm's motion planning is facilitated by the integration of the MoveIt module, renowned for its comprehensive motion planning capabilities, and the OMPL [58] (Open Motion Planning Library), which offers a suite of advanced algorithms for efficient path planning and obstacle avoidance. Upon action completion, the real-time object positions are used to assess task success until manual confirmation or a preset time is reached. In case of failure detection, object and gripper poses are employed for failure reasoning, where the gripper poses are estimated using the attatched QR scan.

# B  Experimental Setup

## B.1  Baseline setup

R3M-DP [13] and DP [7] employ a CNN-based network architecture for its robustness across diverse tasks. These methods are trained on robot demonstrations using default hyper-parameters, robot demonstrations consist of paired observation and action sequences.

To ensure that GraphIRL [24] is trained and tested under the same scenario in the SE setting. GraphIRL is trained in the simulator with corresponding paired robot videos, and the SE results are obtained from the same simulated environments, while UE results are acquired from real-world scenarios under the UE setting. Since the original GraphIRL method struggle to learn the gripper switch information, we additionally provide this information to GraphIRL.

Following Cap [64], the primitives for Cap [64] and demo2code [37] include: move to pos, rotate by quat, set vel, open gripper, close gripper, pick obj, and place at pos. Cap employs natural language instruction directly for reasoning, Demo2code generates code from textual video analysis results provided by GPT-4V for Robotics [33], a video analysis approach for robotics, enabling demo2code to learn from human videos. Specifically, the detailed task analysis results and affordance analysis outcomes from GPT-4V for Robotics are incorporated as contextual information within the textual prompt for demo2code.

## B.2  Real-world experimental setup

Experiments are conducted on Franka Emika [71], employing three RGB-D cameras (ORBBEC Femto Bolt) for environment observation: one at the top right of the table, one at the top left, and one mounted on the robot's wrist. All cameras start recording and return real-time RGB-D observations at a frequency of 30 Hz. All experiments are evaluated on an Intel i7-10700 CPU with an RTX 3090 graphics card.

# C  Details of the long-horizon task designs

The definition of our long-horizon tasks is listed below. For each task, the initial state and subgoals are pre-defined. The whole task is completed if and only if all subgoals are completed in the correct order.

### C.1 Kitchen

- *Make a pie*
  - Initial state: On the table, there is a bowl filled with sauce, a pan containing pie, a brush placed on the shelf, and a microwave.
  - Criteria: Brush the sauce on the pie and put the pie in the microwave to heat up.
  - Subgoals: (1) Grasp the bowl; (2) Pour sauce from the bowl to the pie; (3) Grasp the brush; (4) Spread sauce; (5) Open the microwave; (6) Place the pan in the microwave; (7) Close the microwave; (8) Turn on the microwave.

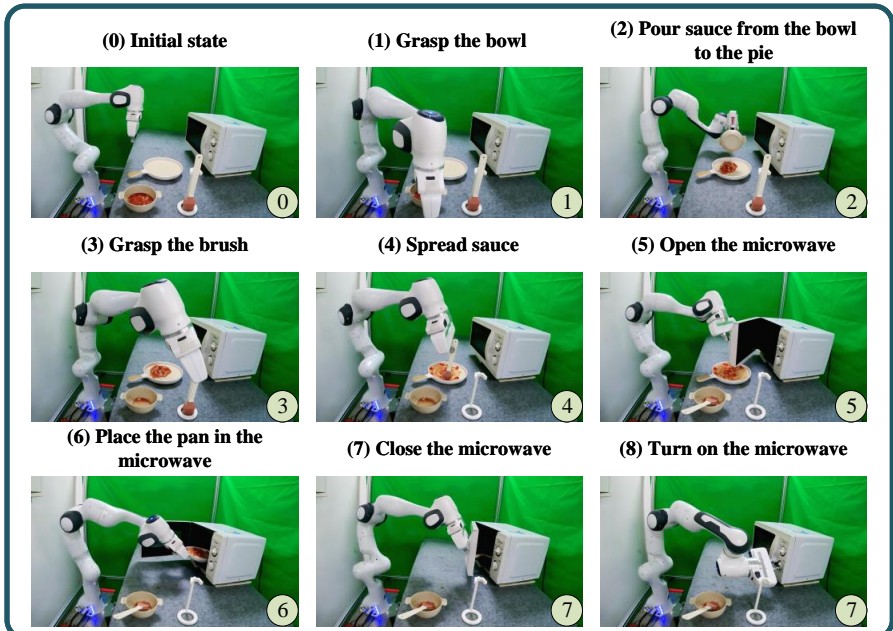

Figure 6: Visualization of the make-a-pie task.

- *Wash pan*
  - Initial state: The pan that needs to be washed is located on the left side of the table, the pan rack and brush are on the right side of the table, and the sink and faucet are in the middle of the table.
  - Criteria: Rinse the pan with water, scrub it with a brush, and place the pan on the rack.
  - Subgoals: (1) Place the pan in the sink; (2)Align the faucet with the pan; (3) Turn on the faucet; (4) Turn off the faucet; (5) Place the pan on the table; (6) Wipe the pan with a brush; (7) Place the pan on the pan rack.

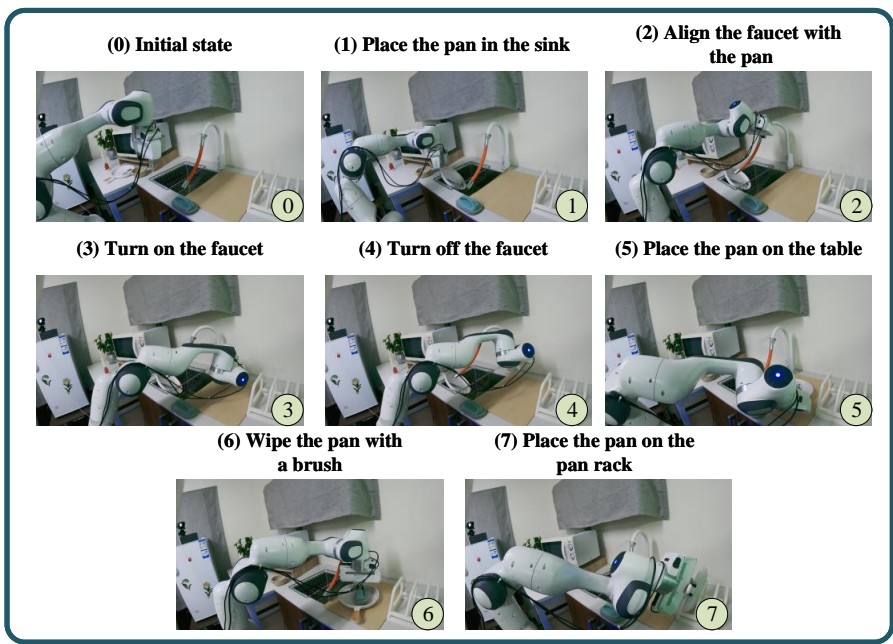

Figure 7: Visualization of the wash-pan task.

- *Make cucumber slices (Make slices)*

    – Initial state: The refrigerator is to the left of the table, the cutting board is on the shelf to the right of the table, next to which is a knife inserted into the knife rack.

    – Criteria: Take the cucumber out of the refrigerator and cut it with a knife.

    – Subgoals: (1) Place the cutting board on the table; (2) Open the refrigerator; (3) Place the cucumber on the cutting board; (4) Close the refrigerator; (5) Remove the knife from the knife rack; (6) Cut the cucumber; (7) Place the knife back on the knife rack.

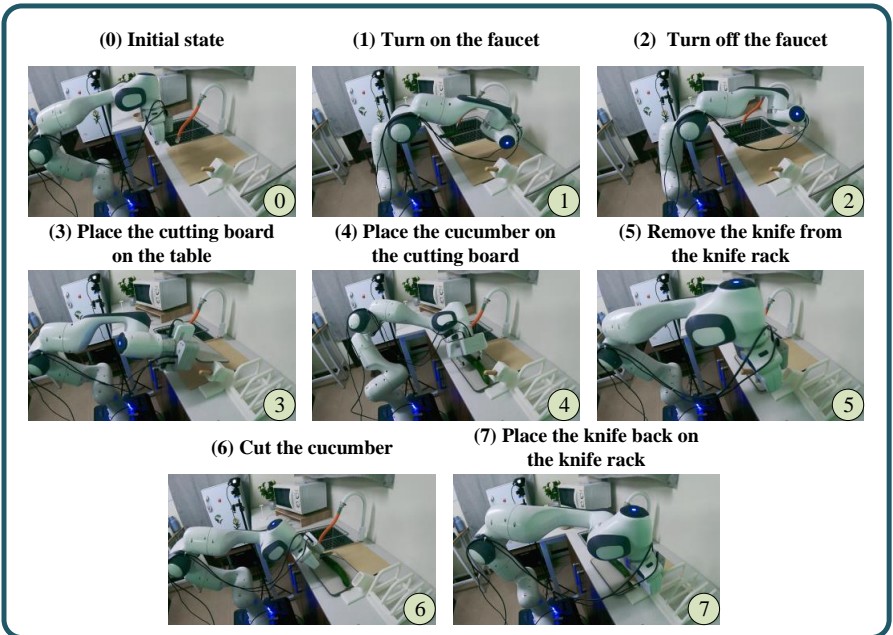

Figure 8: Visualization of the make-cucumber-slices task.

### C.2 Table

- *Make coffee*
  - Initial state: The coffee machine and capsules are placed on the tabletop, with the capsule chamber placed on the cup.
  - Criteria: place the coffee capsule into the capsule chamber, insert it into the coffee machine, place a cup under the coffee machine's dispenser, and finally turn on the coffee machine.
  - Subgoals: (1) Grasp the coffee capsule; (2) Place the coffee capsule in the capsule chamber; (3) Grasp the capsule chamber; (4) Insert the capsule chamber into the coffee machine; (5) Place the cup under the coffee machine's water outlet; (6) Grasp the lever; (7) Turn on the coffee machine.

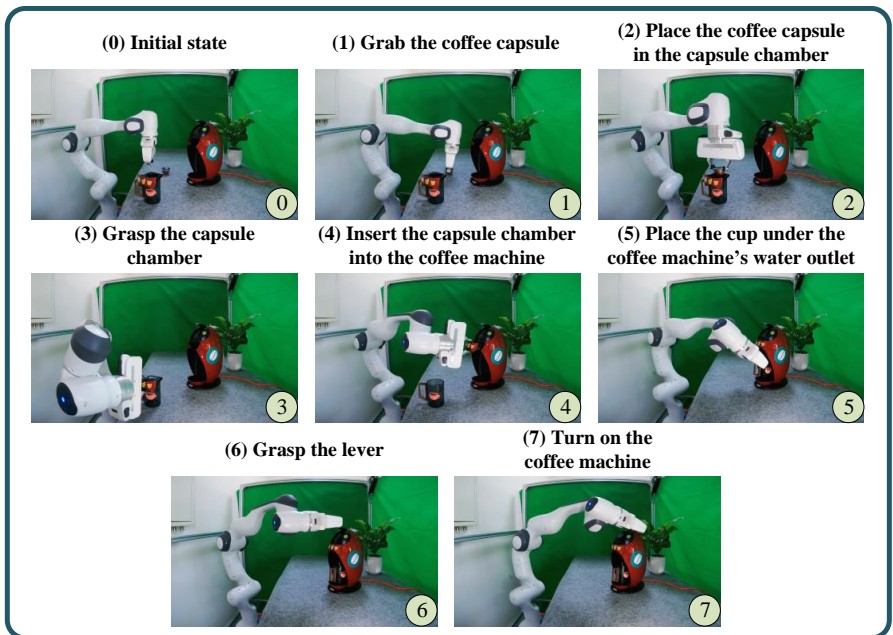

Figure 9: Visualization of the make-coffee task.

- *Clean table*
  - Initial state: Bananas, mangoes, cups, and paint brushes are scattered on the table. In addition, there is a drawer, a plate, and a dust brush on the table.
  - Criteria: Put the fruits (banana and mango) back into the plate, and put the tools (cup and paint brush) back into the drawer, and sweep the tabletop with the dust brush.
  - Subgoals: (1) Place the banana on the plate; (2) Place the mango on the plate; (3) Open the drawer; (4) Place the brush in the drawer; (5) Place the cup in the drawer; (6) Close the drawer; (7) Grasp the brush; (8) Sweep the tabletop.

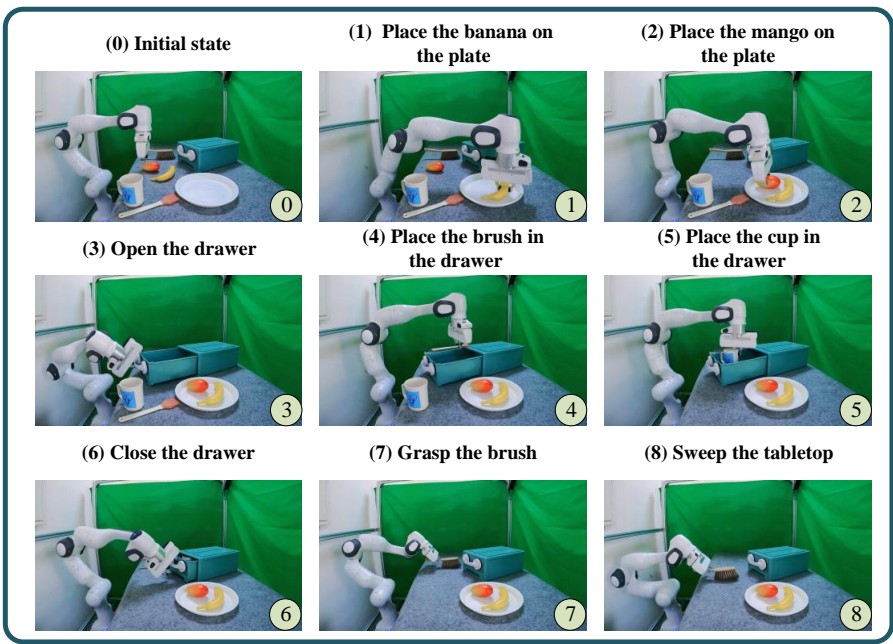

Figure 10: Visualization of the clean-table task.

## C.3 Chemistry Lab

- *Chemistry experiments (Chem. exp.)*
    - Initial state: On the desktop, there are two beakers, two conical flasks, a test tube rack equipped with a test tube, along with a retort stand fitted with a funnel.
    - Subgoals: (1) Place conical flask A under the funnel. (2) Pour the contents of the test tube into beaker A. (3) Pour the contents of beaker A into conical flask B. (4) Pour the contents of beaker B into conical flask B. (5) Shake the mixed solution in conical flask B. (6) Pour the contents of conical flask B into the funnel.

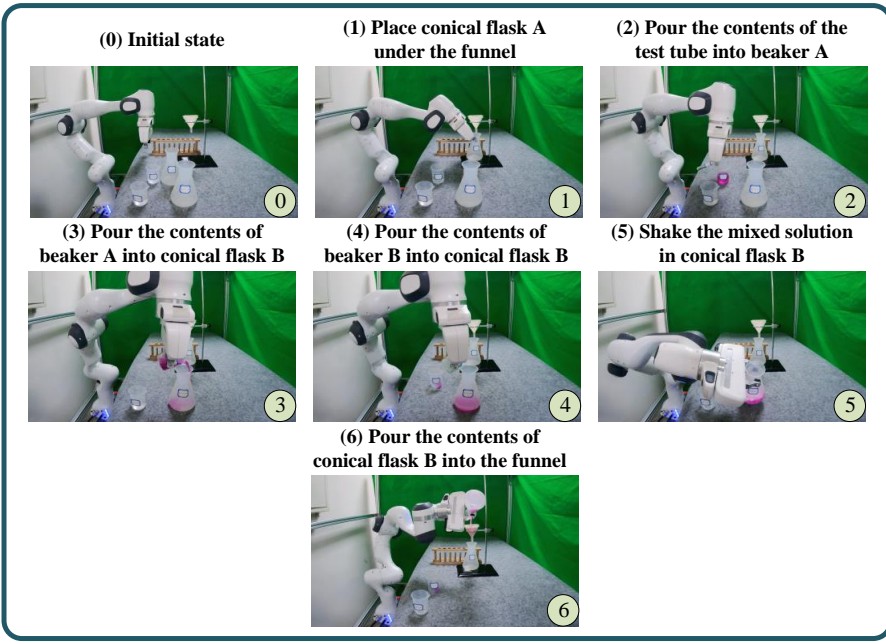

Figure 11: Visualization of the Chemistry Lab task.

# D Visualization of experimental results

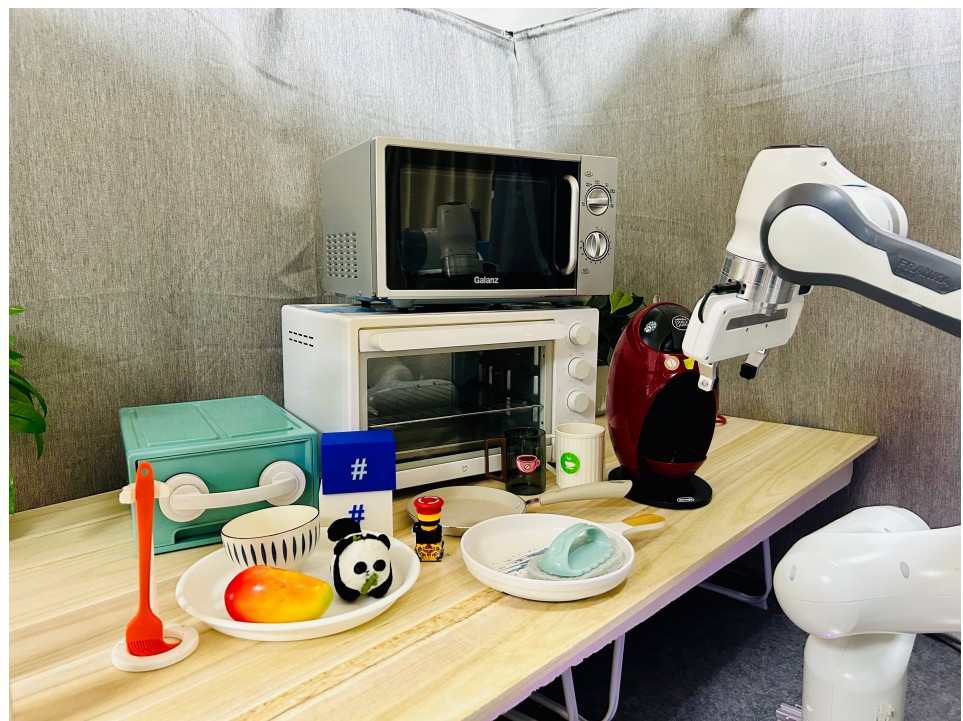
Figure 12: Manipulated objects in SE setting.

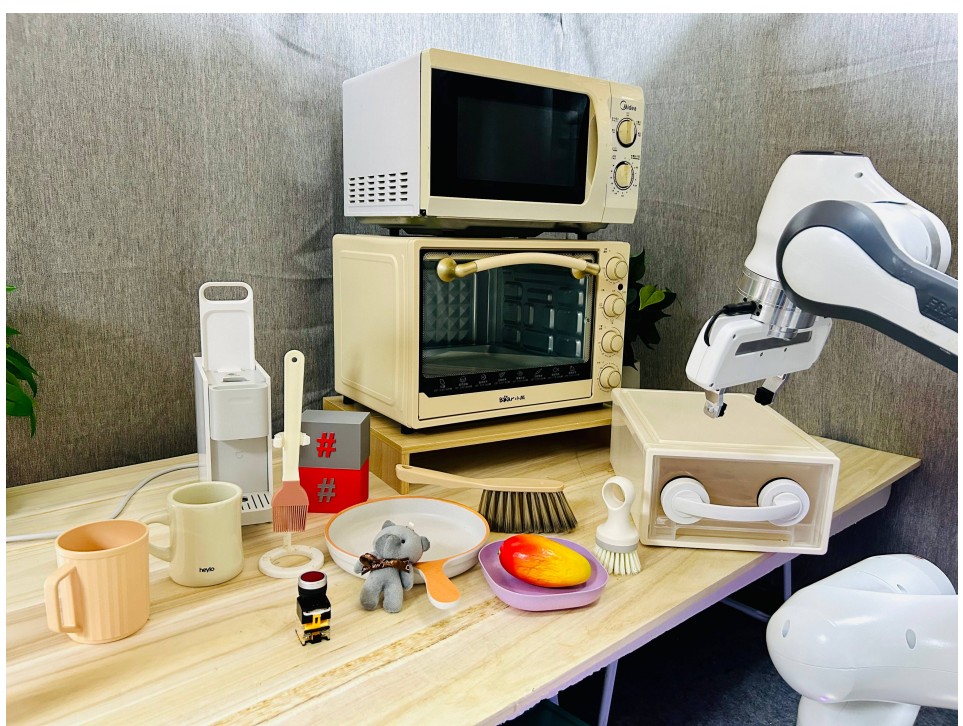
Figure 13: Manipulated objects in US setting.

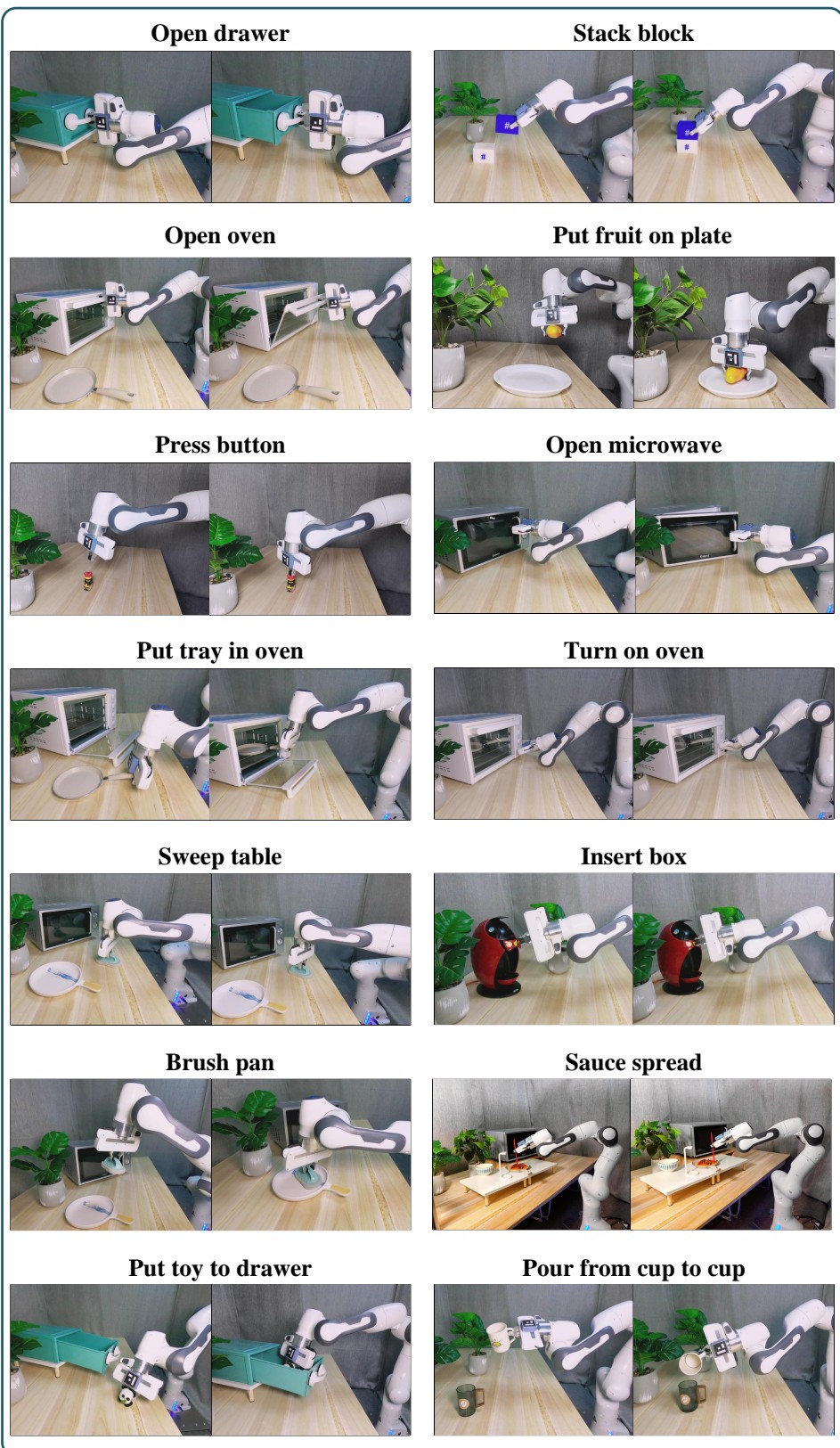

Figure 14: Visualization of manipulation task results in seen environments.

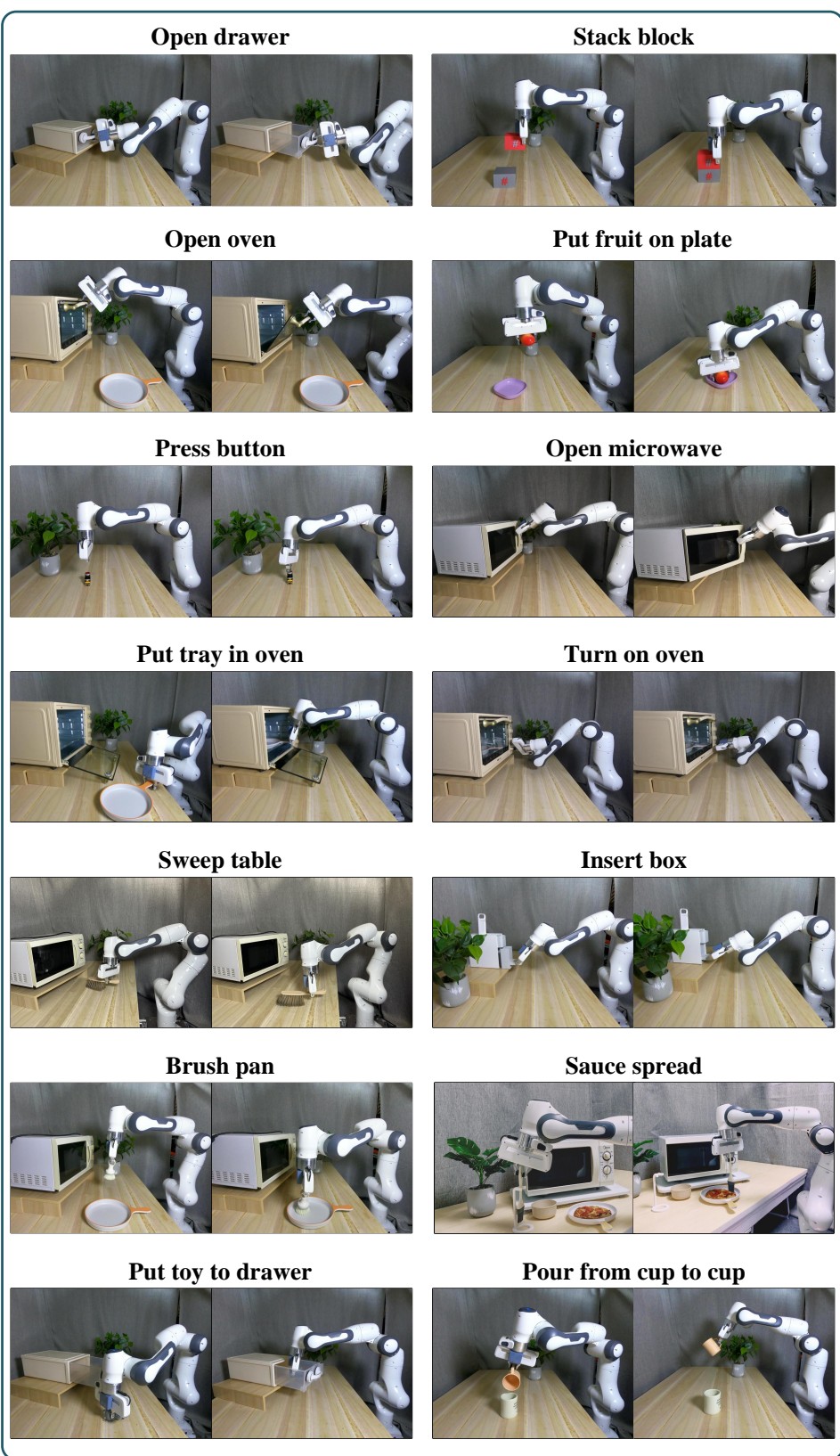

Figure 15: Visualization of manipulation task results in unseen environments.

