# OpenReview forum: "VLMimic: Vision Language Models are Visual Imitation Learner for Fine-grained Actions"
_NeurIPS.cc/2024/Conference — NeurIPS 2024 poster_

### Official Review · Reviewer_ovGy · 2024-07-06

**Soundness:** 4
**Presentation:** 2
**Contribution:** 3
**Rating:** 7
**Confidence:** 4

**Summary:**

This work introduces VLMimic, a framework to acquire robotic skills by imitating from videos.
The system uses VL foundation models to ground human-object interaction videos.
Hierarchical representations are used to learn robotic skills and recorded in knowledge bank.
In unseen environments, the skill adapter iteratively compares to transfer the skill.
In extensive experiments, VLMimic outperforms baselines on the RLBench, as well as comprehensive results in real-world manipulation tasks.

**Strengths:**

It's a strong work.
1. VLMimic not only learns high-level action planning but also learns fine-grained low-level actions, a feature not possessed by past works.
2. The proposed approach looks novel and intuitive.
3. The experiments are impressive. VLMimic outperforms on RLBench simulation. Real-world experiments are conducted to solve long-horizon tasks. Qualitative videos are provided.

**Weaknesses:**

major:
1. Can you predict the system scalability? What's the memory cost of the knowledge bank and the speed cost of iterative comparisons when scaling up? Is it possible to process scaled datasets like Ego-4D?

minor:
1. Some definitions in Line 151 are vague to me. What's the keypoint value? What's the visualized interaction? How do you do the rendering? I suggest some revisions in this part.
2. Inconsistent descriptions: 'high level' and 'high-level'

**Questions:**

I am interested in real-world failure cases of your system, in the cases that can't be corrected by failure reasoning. Which is the main challenge, high-level or low-level?

**Limitations:**

Discussed in Appendix B.

---

> ### Author Rebuttal · Authors · 2024-08-07
>
> **Q1. System scalability.**
>
> We greatly appreciate your constructive suggestions! The computational time and memory requirements are estimated as follows:
>
> **Computational time of iterative comparison**. The majority of computational time in iterative comparison is allocated to awaiting responses from VLMs, accessed via online APIs. In the context of increasing video content, computational efficiency can be maintained by enhancing the rate of parallel API requests, thereby inherently exhibiting favorable scaling properties.
>
> We posit that your primary concern may lie in the time required for skill acquisition, rather than only iterative comparison. Therefore, we estimate the computational time for the entire skill acquisition process.
>
> **Computational time of skill acquisition**. Our approach achieves an average learning time of $443s$ on real-world manipulation tasks, utilizing video sequences of $9s$ on average, captured at 30 Hz. To enhance scalability, we implement the following simple but efficient strategies without compromising accuracy:
>
> - The VLMs are accessed via online APIs, obviating the need for local computational resources. Therefore, we utilize multi-threading to process subsequent videos concurrently with VLM operations, reducing the learning time from $443s$ to $75s$.
> - We decrease the video frame rate from 30 fps to 5 fps, reducing learning time from $75s$ to $12s$.
> - We employ distributed processing by allocating distinct video sequences to separate GPUs, achieving a 9-fold acceleration with 10 GPUs, reducing averaged learning time to $1.4s$.
>
> Consequently, the computational time for learning skills from the 120-hour subset of Ego4D is estimated to be $120 \times 1.4 / 9 = 18.6h$. Furthermore, acceleration can be further achieved by eliminating invalid video clips.
>
> **Memory cost**. The average memory allocation per skill is approximately 5MB. Given that the average temporal duration of skills is $9s$, the estimated number of skills in Ego-4D is $120 \times 60 \times 60 / 9 = 48000$, and the memory cost is $48000 \times 5/1024 = 234.3GB$.
>
> Overall, our method exhibits promising scaling properties, with an estimated computational time of less than $19h$ and a required storage capacity below $240 GB$.
>
> **Q2. Some definitions in Line 151.**
>
> Thanks for your valuable feedback! We will respond to your questions sequentially.
>
> **Keypoint value**. keypoint values denote critical interaction information. In the grasping phase, the keypoints are represented by the grasp poses. In the manipulation phase, keypoints represent a compressed object-centric trajectory, derived through a uniform sampling of 10 points along the estimated trajectory.
>
> **Visualized interactions**. Visualized interactions comprise a series of images that illustrate object grasping poses and manipulation trajectories. For the grasping phase, the visualized interactions comprise the object and multiple grippers. These grippers are displayed according to the object-centric grasp poses. For the manipulation phase, the visualized interactions consist of the master object and the estimated master object-centric motion trajectories of slave objects. We recall that the agent employs a slave object to interact with a master object in the manipulation phase.
>
> **Rendering pipeline**. In the grasping phase, we position the object at the origin of the 3D coordinate system, then simplified grippers are placed by the estimated object-centric grasping poses. Following this, this 3D scene undergoes projection onto 2D images along the X, Y, and Z directions, obtaining visualized interactions for the grasping phase. For the manipulation phase, the master object is positioned at the origin of the 3D coordinate system. Subsequently, the 3D trajectory is generated in concordance with the estimated master object-centric slave pose trajectory. The visualized interactions for the manipulation phase are obtained by projecting this scene into 2D images along the X, Y, and Z directions.
>
> **Q3. Inconsistent descriptions.**
>
> Thanks for your valuable feedback! We will make the revisions to ensure consistency and conduct a comprehensive review of the entire article to identify and rectify any potential inconsistencies.
>
>
> **Q4. Real-world failure cases.**
>
> Insightful question! Figure 6 is presented in the attached PDF to illustrate situations challenging to resolve through failure reasoning, including:
> - The task execution may exceed the hardware limitations of the physical robot, inducing **inverse kinematics (IK) errors**.
> - Incomplete environmental perception increases the risk of **obstacle collisions**, leading to task failure.
>
> The training datasets for VLMs exhibit a significant lack of data related to robot dynamics. Consequently, these models exhibit a limited capacity for error analysis and struggle to infer correction strategies when confronted with these failures.
>
>
> **Q5. Which is the main challenge?**
>
> Excellent question! We believe that **low-level challenges are more significant**, due to several factors:
> - **Low-level planning is more complex compared to high-level planning.** (1) Low-level planning requires **detailed operational specifications**, like grasping poses and motion trajectories, while high-level planning only focuses on overall task objectives. (2) Low-level planning must account for **diverse physical constraints**, such as hinge mechanisms of microwave doors. Conversely, high-level planning abstracts these physical interactions. (3) Low-level planning needs to **handle uncertainties** such as sensor noise, while high-level planning assumes an idealized environment.
> - **LLMs/VLMs exhibit superior proficiency in high-level planning**. High-level planning focuses on abstract strategic formulation, which aligns with the training corpus of VLMs. Conversely, low-level planning necessitates reasoning about fine-grained interactions, which is scarce in VLMs' training corpora, resulting in a capability deficit.

---

> ### Author Response · Authors · 2024-08-09
>
> Dear Reviewer ovGy,
>
> Thank you again for reviewing our manuscript. We have tried our best to address your questions (see our rebuttal in the top-level comment and above). Please kindly let us know if you have any follow-up questions or areas needing further clarification. Your insights are valuable to us, and we stand ready to provide any additional information that could be helpful.

---

> > ### Comment · Reviewer_ovGy · 2024-08-12
> >
> > The reviewer thanks authors' responses. My concerns are addressed. I keep my original rating.

---

> > > ### Author Response · Authors · 2024-08-12
> > > **Thanks for your feedback!**
> > >
> > > Thanks for considering our responses and recommending acceptance! We will update our revised paper according to our discussions. Thanks again for your insightful and constructive suggestions for improving paper quality. We are happy to address any further questions or concerns about the work.

---

### Official Review · Reviewer_aTuD · 2024-07-09

**Soundness:** 3
**Presentation:** 4
**Contribution:** 2
**Rating:** 6
**Confidence:** 5

**Summary:**

The paper proposes a system that can perform imitation learning based on human demonstration videos. The authors design a system using many pre-trained components like VLMs and hand and object tracker and 3D reconstruction and pose estimation tools. They show that by combining these existing tools they can achieve compelling success rates on multiple robots.

**Strengths:**

1. Impressive performance across real robots and multiple tasks.
2. Ablation studies showing importance of different components.
3. Detailed analysis of runtime of each component.

**Weaknesses:**

1. The paper and the abstract give a lot of credit to "VLMs". However there are a lot of other learnt modules like SAM-Track, BundleSDF and FoundationPose in the system. A lot of hand-designed priors like 3D projection, grasp prediction etc are also baked into the system. These should also be given due credit along with VLMs.

2. The paper seems to be more of a robotics systems paper where many modules are connected to perform a task than a machine learning paper.

**Questions:**

1. Which models are actually used for the system? The authors mention multiple models many times in the paper: VFM [34; 35; 36] , SAM-Track [37; 38; 39; 40; 41]. Which of these are actually used?
2. What is the knowledge bank used in the paper? Examples of how it is constructed and accessed?
3. How many times is each method run during evaluation? How much scene randomization is done between these episodes and how different are they from the training demonstrations?

**Limitations:**

Yes

---

> ### Author Rebuttal · Authors · 2024-08-07
>
> **Q1. Give a lot of credit to VLMs**
>
> Thanks for your valuable suggestions! We give credit to VLMs since the majority of **other foundation models serve as data generators** within the human-object interaction grounding module, providing motion data for VLMs to analyze, while **the acquisition and generalization of skills are mainly achieved through VLMs**, enabling the acquisition of robust and generalizable skills from limited demonstrations.
>
> **We acknowledge the important role of these foundation models in our framework**. Specifically, we will revise our manuscript to emphasize:
> - The utilization of foundation models for direct learning of fine-grained actions.
> - The critical function of foundation models in estimating object-centric actions for VLM analysis.
>
> **Q2. The paper seems to be more of a robotics systems paper.**
>
> Thanks for your feedback! We have indeed submitted our paper on the topic of robotics. Our research primarily focuses on **leveraging machine learning approaches to enhance embodied intelligence in robotic systems**.
>
> Our paper presents a novel approach that leverages the capabilities of VLMs, aiming to **address a crucial challenge in machine learning: the acquisition of robust, generalizable skills from limited demonstrations**. Furthermore, our study represents **a pioneering achievement of VLMs in fine-grained motion skill learning**, demonstrating its ability to comprehend real-world interactions. To achieve this, we propose a novel framework incorporating hierarchical constraint representations and an iterative comparison methodology to enhance the reasoning capabilities of VLMs. Our approach leverages the advanced capabilities of foundation models, aligning with the prevailing paradigm in machine learning research that seeks to harness these advanced models for the augmentation of diverse task performance.
>
> Finally, we would like to clarify that the implemented models, primarily within the human-object interaction grounding module, function as a data generator. **This multi-module approach for robust, automated data generation is consistent with practices in many machine learning papers.**
>
> **Q3. Which models are used for the system?**
>
> Thanks for your valuable feedback! The cited VFM models are capable of executing our tasks, with the employed model detailed in the Appendix's Implementation Details section. To enhance clarity, we will append a statement to the mentioned paragraph specifying that Tokenize Anything [5] is utilized in our experiments.
>
> Concerning SAM-Track [6], we adhere to the authors' guidelines on GitHub and cite these works. To reduce ambiguity, we will exclusively cite the SAM-Track paper.
>
> [5] Pan T, Tang L, Wang X, et al. Tokenize anything via prompting[J]. arXiv preprint arXiv:2312.09128, 2023.
>
> [6] Cheng Y, Li L, Xu Y, et al. Segment and track anything[J]. arXiv preprint arXiv:2305.06558, 2023.
>
> **Q4. Knowledge bank and its examples.**
>
> The knowledge bank functions as a repository for archiving both high-level planning and low-level skill insights. Upon encountering novel environments, relevant knowledge is retrieved from this repository. The intuitive example is presented in Figure 8 of the attached PDF.
>
> **Construction.** A knowledge bank is established to archive both high-level planning and low-level skill insights, storing knowledge with key-value pairs. High-level planning knowledge is indexed using task description as keys, paired with the consequent action sequence as values. For low-level skill knowledge, keys are constituted by the object images and subtask description, and values comprise reconstructed objects, as well as semantic constraints and geometric constraints representing learned skills.
>
> **Access.** (1) High-level knowledge retrieval: The text encoder [7] is leveraged to obtain the similarity between the queried task description and the task descriptions associated with stored demonstrations. Then, the demonstration with the highest similarity is retrieved. (2) Low-level skill knowledge retrieval: Based on the queried subtask description, we leverage the text encoder [7] to select $N_{t}=2$ demonstrations with the highest similarity between their associated subtasks and the query. Subsequently, from the selected demonstrations, those exhibiting the highest degree of object similarity are retrieved. The image similarities based on the CLIP model [8] are calculated for both master and slave objects between the current scenarios and demonstrations. The average of these similarities is utilized as the similarity metric.
>
> [7] Liu Y, et al. Roberta: A robustly optimized Bert pretraining approach[J]. arXiv preprint arXiv:1907.11692, 2019.
>
> [8] Radford A, et al. Learning transferable visual models from natural language supervision[C]//ICML. PMLR, 2021: 8748-8763.
>
> **Q5. Number of evaluations and scene randomization.**
>
> **Number of evaluations**. The success rate in the RLBench simulation environment is calculated as the mean of 100 independent trials. For real-world experiments, following Voxposer [9], the success rate is determined by computing the average outcome of 10 discrete tests.
>
> **Scene randomization**. The position and orientation of objects are randomly determined in both the RLBench simulation and real-world experimental setups. Specifically, the spatial coordinates of objects are randomly sampled within the operational range of the robotic manipulator on the task workspace. Concurrently, the rotational orientation of objects is randomly selected from a predetermined set of feasible angular configurations (such as $[-\pi/2,\pi/2]$ for the $Z$-axis of the microwave). Additionally, within the RLBench simulation framework, the color attributes of the objects are subjected to probabilistic sampling.
>
> [9] Huang W, Wang C, Zhang R, et al. VoxPoser: Composable 3D Value Maps for Robotic Manipulation with Language Models[C]//NeurIPS 2023 Foundation Models for Decision Making Workshop.

---

> ### Author Response · Authors · 2024-08-09
>
> Dear Reviewer aTuD,
>
> Thank you again for reviewing our manuscript. We have tried our best to address your questions (see our rebuttal in the top-level comment and above). Please kindly let us know if you have any follow-up questions or areas needing further clarification. Your insights are valuable to us, and we stand ready to provide any additional information that could be helpful.

---

> > ### Comment · Reviewer_aTuD · 2024-08-10
> >
> > Thanks for the clarifications. I have updated the score to weak accept. Please include the answers in the final version of the paper.

---

> > > ### Author Response · Authors · 2024-08-11
> > > **Thanks for your feedback!**
> > >
> > > Thanks for considering our responses and raising the score! We will update our revised paper according to our discussions. Thanks again for your insightful and constructive suggestions for improving paper quality. We are happy to address any further questions or concerns about the work.

---

### Official Review · Reviewer_xpmL · 2024-07-10

**Soundness:** 3
**Presentation:** 3
**Contribution:** 2
**Rating:** 6
**Confidence:** 4

**Summary:**

This paper introduces a novel paradigm named VLMimic, which employs Vision-Language Models (VLMs) and multiple vision tools to ground object-centric information from demonstrations into fine-grained actionable skill sets. VLMimic exhibits a remarkable success rate in manipulation tasks and demonstrates significant generalization capabilities, learning effectively from a minimal number of human and robot videos.

**Strengths:**

1. The proposed method performs well even with very few demonstrations (less than 5 videos).
2. The evaluations are substantial, encompassing both simulation and real-world experiments, with diverse tasks conducted and generalization of multiple aspects (position, intervention, cross-embodiment and cross-task) verified.
3. The writing in general is well-organized and easy to follow.

**Weaknesses:**

The proposed object-centric manipulation pipeline encompasses multiple foundation models / vision tools to generate various intermediate representations (object detection results, object 6D poses, hand mask and trajectories, and task description from VLMs). This integration relies heavily on human prior knowledge to string all these models, rendering the process lengthy and complex.  Moreover, this approach may only be adaptable to a limited set of interaction objects (for example, whose 'affordances',  or 'object properties' as proposed in the paper, can be easily defined and recognized).


*[Minor]* According to line167-168, the semantic constraints are identified based on both visual interaction $I_{v}$ and sub-task description $T_{\tau}$. But only  $I_{v}$ is incorporated in Equation 3.

**Questions:**

1. In the iterative comparison process, how does VLMimic determine which object, from the extensive knowledge bank of diverse objects, is utilized as the reference for guiding the adaptation of grasping and manipulation constraints?

2. Has the potential for cumulative error in the proposed modular design been taken into account? Specifically, is there a risk that an error at any stage of the process could ultimately lead to the failure of the manipulation task?

**Limitations:**

Possible limitations are discussed in the paper.

---

> ### Author Rebuttal · Authors · 2024-08-06
>
> **Q1. The proposed pipeline encompasses multiple foundation models/vision tools.**
>
> Thanks for your feedback! We hope to address your concerns through the following two points.
>
> (1) **Our VLMimic demonstrates strong robustness against the cumulative error.** Motivated by your insightful feedback, we conduct an evaluation of the resilience of VLMimic to cumulative errors through perturbation introduction at various stages. The empirical results substantiate the robust performance of our framework. Details are presented in section Q5.
>
> |VLMimic|Interaction grounding errors|Knowledge retrieval errors|Execution errors|
> |-|-|-|-|
> |$0.76 (\pm 0.17)$|$0.68 (\pm 0.21)$|$0.64 (\pm 0.21)$|$0.73 (\pm 0.17)$|
>
> (2) **Foundation models/vision tools are primarily utilized in the data generation phase, which can facilitate efficient skill learning without compromising robustness.** The implemented foundation models are primarily distributed in the human-object interaction grounding module, which functions as a data generator. This automated data generation approach substantially mitigates the need for human data annotation. Furthermore, selective manual annotation of a minimal subset of erroneous data can further enhance data quality. Therefore, the integration of foundation models enables accelerated skill acquisition paradigms while preserving the robustness of the learning process.
>
> **Q2. Learned interactions are confined to the easily defined affordances or object properties.**
>
> Our extensive experiments demonstrate the method's broad versatility across diverse objects and interactions:
>
> **Object diversity**: The introduced object properties primarily comprise bounding boxes, universally applicable to all objects. The experiments encompass **42 distinct object classes**, ranging from household objects to laboratory equipment, illustrating adaptability to varied objects.
>
> **Affordance complexity**: Our method demonstrates proficiency in **62 low-level skills**, including complex interactions like capsule insertion and liquid transfer, showcasing its ability to learn diverse affordances. Moreover, we extend our evaluation to include more challenging tasks, **demonstrating strong performance on FMB [1], which demands precise manipulation**. FMB evaluates complex robotic manipulation tasks encompassing the grasping and reorienting that utilizes the fixture, and inserting objects. Our method achieves superior performance to DP, requiring only five human videos. An example is illustrated in Figure 1 of the attached PDF.
>
> |Methods|Type of demos|Num of demos|Overall|Round|Rectangle|Oval|Hexagon|Arch|Star|Double Square|Square Circle|Triple Prong|
> |-|-|-|-|-|-|-|-|-|-|-|-|-|
> |DP|Obs-act|100|$0.47 (\pm 0.16)$|0.70|0.60|0.40|0.40|0.30|0.70|0.40|0.30|0.40|
> |Ours|Video|5|$0.49 (\pm 0.13)$|0.80|0.50|0.40|0.50|0.40|0.50|0.50|0.40|0.40|
>
> [1] Luo J, et al. FMB: a Functional Manipulation Benchmark for Generalizable Robotic Learning[C]//CoRL 2023 Workshop on LEAP.
>
> **Q3. Equation 3.**
>
> Thanks for your feedback! We will modify the equation according to your feedback and review all parts of the article to address any similar issues.
>
> **Q4. How does VLMimic determine which object is utilized as the reference?**
>
> The retrieval example is presented in the lower part of Figure 8 within the attached PDF: Based on the queried subtask description, we leverage the text encoder [2] to select $N_{t}=2$ demonstrations with the highest similarity between their associated subtask descriptions and the query. Subsequently, from the selected demonstrations, those exhibiting the highest degree of object similarity are retrieved. The image similarities based on the CLIP model [3] are calculated for both master and slave objects between the current scenarios and demonstrations. The average of these similarities is utilized as the similarity metric.
>
> [2] Liu Y, Ott M, Goyal N, et al. Roberta: A robustly optimized Bert pretraining approach[J]. arXiv preprint arXiv:1907.11692, 2019.
>
> [3] Radford A, Kim J W, Hallacy C, et al. Learning transferable visual models from natural language supervision[C]//ICML. PMLR, 2021: 8748-8763.
>
> **Q5. Cumulative error.**
>
> Thanks for your constructive suggestion! **Our VLMimic demonstrates strong robustness against the cumulative error.** Validation is conducted on the simulation manipulation tasks to assess our system's resilience to cumulative errors. We inject errors at the input of skill learning, adapting, and execution phases, corresponding to interaction grounding errors, knowledge retrieval errors, and execution errors, respectively:
>
> - **Interaction grounding errors**. Gaussian noise ($\sigma = 5cm$ for the position, $\sigma = 5^{\circ}$ for rotation) is applied to pose estimation results from human videos, referring to the Megapose evaluation metrics [4]. These metrics quantify prediction accuracy based on the percentage of estimates within $5^{\circ}$ rotational and $5cm$ translational thresholds from the ground truth.
> - **Knowledge retrieval errors**. A knowledge base is constructed using RLBench experimental learning data. During testing, relevant knowledge is retrieved from this knowledge base. To introduce errors, the most similar matches among all 12 matches are deliberately omitted.
> - **Execution errors**.  Gaussian noise ($\sigma = 5cm$ for position, $\sigma = 5^{\circ}$ for rotation) is applied to object pose estimation results.
>
> |VLMimic|Interaction grounding errors|Knowledge retrieval errors|Execution errors|
> |-|-|-|-|
> |$0.76 (\pm 0.17)$|$0.68 (\pm 0.21)$|$0.64 (\pm 0.21)$|$0.73 (\pm 0.17)$|
>
> Furthermore, several illustrative examples of robustness against perturbation and failure reasoning are provided in Appendix E and K of the paper.
>
> [4] Labbé Y, Manuelli L, Mousavian A, et al. MegaPose: 6D Pose Estimation of Novel Objects via Render & Compare[C]//CoRL 2022-Conference on Robot Learning. 2022.

---

> ### Author Response · Authors · 2024-08-09
>
> Dear Reviewer xpmL,
>
> Thank you again for reviewing our manuscript. We have tried our best to address your questions (see our rebuttal in the top-level comment and above). Please kindly let us know if you have any follow-up questions or areas needing further clarification. Your insights are valuable to us, and we stand ready to provide any additional information that could be helpful.

---

> ### Comment · Reviewer_xpmL · 2024-08-11
>
> Thanks for the thorough rebuttal provided by the authors. Most of my concerns were addressed, but the method's ad-hoc design may still limit its impact to broader audiences. I’m maintaining my initial score.

---

> > ### Author Response · Authors · 2024-08-11
> > **Thanks for your feedback!**
> >
> > Thanks for considering our responses and recommending acceptance! We will update our revised paper according to our discussions. Thanks again for your insightful and constructive suggestions for improving paper quality. We are happy to address any further questions or concerns about the work.

---

### Official Review · Reviewer_Y6C5 · 2024-07-17

**Soundness:** 3
**Presentation:** 3
**Contribution:** 2
**Rating:** 6
**Confidence:** 3

**Summary:**

This paper proposes a method to learn a policy from human videos utilizing advances in vision-language models. The authors first parse human videos into several keyframes, then detect and track hand-object interactions. From the segmented interactions, they learn low-level actions by following the 3D hand-object trajectories in each segment. Additionally, the authors propose an iterative comparison stage to adapt to new scenes or objects. The method achieves better performance compared to other visual imitation learning algorithms in both simulation and real-world evaluations.

**Strengths:**

This paper proposes a novel usage of VLM for imitation learning. The perception module is well-designed, and the ability to learn skills from perceived trajectories is particularly interesting.

The paper presents comprehensive evaluations in both simulation and the real world. It also includes detailed ablation experiments on different design choices of the proposed system.

It demonstrates superior performance over previous methods, especially on unseen objects and tasks.

**Weaknesses:**

While this paper is well-written and achieves good performance, I have a few concerns regarding some claims and evaluations of the proposed method.

First, the paper focuses on fine-grained action imitations. However, I’m not convinced that the skills demonstrated in the paper are fine-grained or distinct from previous work that uses VLM for planning. Specifically, the skills in this paper primarily involve grasping and moving along certain trajectories, without complex interactions (actions requiring precise manipulation such as insertion or grasping that utilizes the environment).

On the other hand, similar skills can also be learned using the 'VLM as Planner' framework. For example, the 'Code as Policy' approach can perform tasks such as writing with pens. This skill is quite similar to the 'Spread sauce' motion described. Therefore, the claim that 'Fine-grained motions' cannot be learned in the 'VLM as Planner' framework needs further explanation.

The proposed method seems to categorize motion skills as 'grasping' and 'manipulation'. If the point is that 'VLM as Planner' requires each skill to be defined manually while the proposed method can automate this process, it would be beneficial to clarify that the proposed method still defines motion skills, but in a broader or more generalizable manner.

I think the performance of the baseline imitation learning algorithms is surprisingly low. Given 100 demonstrations, R3M-DP and DP should be quite successful, at least in simpler tasks such as Reach Target. This assessment is based on the original paper where they achieve a high success rate in more difficult tasks, as well as my empirical experience. Could you clarify whether the evaluation protocol was changed to make the task much harder?

**Questions:**

The proposed system is rather complicated. While I appreciate the authors' efforts in assembling such a complex system, it is honestly a bit difficult to understand. I have detailed a few technical questions below.
In the grounding module:
* How are the keyframes obtained?
* How does the language model identify the important object to detect (e.g., pie, plate, brush, and bowl)?
* How are candidate action primitives (such as grasping and manipulation) determined?
For skill learning:
* Can you provide some examples of the Multiple-choice QA?
* How are the keypoints obtained?

I appreciate the evaluation on unseen scenes and objects. However, how similar do the camera viewing angles need to be? I assume there must be some degree of robustness because the perception operates in 3D, but is there any evaluation of this aspect?

As mentioned in the previous section, the results for R3M-DP and DP are surprisingly low. Have you tested the implementation on other tasks (such as those reported in the original paper) to verify the correctness?

**Limitations:**

The authors report limitations in the supplementary material. The limitations section is comprehensive, though I would suggest moving it to the main text.

---

> ### Author Rebuttal · Authors · 2024-08-06
>
> **Q1. The tasks lack complex interactions.**
>
> Thanks for your valuable feedback! We extend our evaluation to include complex tasks, and **demonstrate strong performance on FMB [1], which demands precise manipulation**. FMB evaluates complex robotic manipulation tasks encompassing the grasping and reorienting that utilizes the fixture, and inserting objects. Our method outperforms comparison methods, requiring only five human videos. An example is illustrated in Figure 1 of the attached PDF.
>
> |Methods|Type of demos|Num of demos|Overall|Round|Rectangle|Oval|Hexagon|Arch|Star|Double Square|Square Circle|Triple Prong|
> |-|-|-|-|-|-|-|-|-|-|-|-|-|
> |DP|Obs-act|100|$0.47(\pm0.16)$|0.7|0.6|0.4|0.4|0.3|0.7|0.4|0.3|0.4|
> |CaP|Template|5|$0.11(\pm0.19)$|0.6|0.2|0.1|0.0|0.0|0.1|0.0|0.0|0.0|
> |Ours|Video|5|$0.49(\pm0.13)$|0.8|0.5|0.4|0.5|0.4|0.5|0.5|0.4|0.4|
>
> [1] Luo J, et al. FMB: a Functional Manipulation Benchmark for Generalizable Robotic Learning[C]//CoRL 2023 Workshop on LEAP.
>
> **Q2. Similar skills can also be learned using the 'VLM as Planner' framework.**
>
> Sorry for the confusion. We would like to clarify that **while methods utilizing VLMs as planners can perform similar tasks, fine-grained motion skills are not learned by themselves**. As shown in Figure 2 of the attached PDF,  (a) the motion skill in the drawing experiments encompasses fine-grained control to ensure continuous contact and trajectory adherence of the pen on the whiteboard surface, this skill is implemented via the **human-written function** `draw(pts_2d)`. In contrast, (b) these fine-grained motion skills are autonomously acquired by our VLMimic.
>
> **Q3. The proposed method still defines motion skills.**
>
> Thanks for your valuable feedback! We will clarify that VLMimic defines skills in a broader manner. Furthermore, it is crucial to emphasize that **the motion primitives for 'VLM as planner' methods encompass specific skill implementations**, necessitating manual function coding, or specific model training. In contrast, **our approach merely categorizes skills into grasping and manipulation, without additional human efforts**, therefore significantly augmenting the task learning capabilities for robotic systems.
>
> **Q4. Performance of Diffusion policy.**
>
> **In RLBench**, we follow the commonly utilized mutli-task training paradigm and implement the task-conditioned DP approach, however, DP fails to manage different skills and generalize to test examples. Following your suggestions, we conduct single-task experiments, wherein each DP model is constrained to execute a singular task. Despite these constraints, VLMimic still exhibits superior performance, requiring only 5 human videos.
>
> |Methods|DP|Ours|
> |-|-|-|
> |Overall|$0.66(\pm0.28)$|${0.80(\pm0.16)}$|
>
> |Methods|Type of demos|Num of demos|Reach target|Take lid off saucepan|Pick up cup|Toilet seat up|Open box|Open door|
> |-|-|-|-|-|-|-|-|-|
> |DP|Obs-act|100|1.00|1.00|0.85|0.77|0.57|0.67|
> |Ours|Video|5|1.00|0.98|0.85|0.79|0.76|0.92|
>
> |Methods|Type of demos|Num of demos|Meat off grill|Open drawer|Open grill|Open microwave|Open oven|Knife on board|
> |-|-|-|-|-|-|-|-|-|
> |DP|Obs-act|100|0.83|0.79|0.73|0.21|0.10|0.41|
> |Ours|Video|5|0.87|0.77|0.85|0.47|0.50|0.79|
>
> **In the real-world experiments**, we follow the evaluation protocol in the original paper of DP and discover two primary failure factors: (1) DP exceeds robot hardware limits, inducing inverse kinematics errors. (2) DP exhibits safety-violating behaviors, such as hitting the ground.
>
> **Q5. How are the keyframes obtained?**
>
> Keyframes are extracted by sampling video frames every three seconds. In instances where the extracted keyframes are numbered fewer than five, we uniformly sample five keyframes.
>
> **Q6. Important object identification.**
>
> As shown in Figure 3 of the attached PDF, keyframes are grouped in fives and concatenated into consolidated images. These images prompt VLMs to generate task descriptions. VLMs then analyze individual keyframes based on task descriptions, selecting important objects from detection results.
>
> **Q7. Candidate action primitives determination.**
>
> Keyframes are extracted from subtask video segments. These keyframes, along with interacting objects within the subtask, are input to VLMs to distinguish between the grasping and manipulation phases. The interacting objects are the two objects satisfy ${d}^{t-1}>\epsilon\wedge{d}^{t}<\epsilon$, where $d$ denotes the distance between two objects. Examples are provided in Figure 9 of the attached PDF.
>
> **Q8. Examples of the Multiple-choice QA.**
>
> As illustrated in Figure 4 of the attached PDF, VLMimic demonstrates proficiency in inferring grasping constraints.
>
> **Q9. How are the keypoints obtained?**
>
> In the grasping phase, the keypoints are represented by the grasp poses.
> In the manipulation phase, keypoints are derived through a uniform sampling of 10 points along the estimated trajectory.
>
> **Q10. Viewpoint variance.**
>
> Thanks for your constructive suggestion! Experiments are conducted in real-world unseen environments, utilizing distinct viewpoints, as shown in Figure 5 of the attached PDF. Results demonstrate the resilience of VLMimic to viewpoint changes.
>
> |Methods|Viewpoint 1|Viewpoint 2|Viewpoint 3|Viewpoint 4|
> |-|-|-|-|-|
> |Ours|$0.71(\pm0.15)$|$0.67 (\pm 0.16)$|$0.70 (\pm 0.15)$|$0.64 (\pm 0.17)$|
>
> **Q11. Diffusion policy on similar tasks.**
>
> DP performs best in 'Lift' and 'Can' tasks. Tests on similar RLBench tasks show success rates in single-task training align with the original study's findings.
>
> |Methods|Type of demos|Num of demos|Overall|Reach target|Take lid off saucepan|press button|Meat off grill|
> |-|-|-|-|-|-|-|-|
> |DP|Obs-act|100|$0.93(\pm0.08)$|1.00|1.00|0.91|0.83|
>
> Real-world experiments reveal decreased success rates, primarily due to inverse kinematics errors and safety violations, as mentioned in Q4.
>
> |Methods|Type of demos|Num of demos|Overall|Reach target|Lift|Can|
> |-|-|-|-|-|-|-|
> |DP|Obs-act|100|$0.81(\pm0.08)$|0.90|0.80|0.75|

---

> ### Author Response · Authors · 2024-08-09
>
> Dear Reviewer Y6C5,
>
> Thank you again for reviewing our manuscript. We have tried our best to address your questions (see our rebuttal in the top-level comment and above). Please kindly let us know if you have any follow-up questions or areas needing further clarification. Your insights are valuable to us, and we stand ready to provide any additional information that could be helpful.

---

> ### Comment · Area_Chair_VvHz · 2024-08-12
> **To reviewer Y6C5: Please respond to rebuttal**
>
> Hi reviewer Y6C5,
>
> Thank you for your initial review. Please kindly respond to the rebuttal posted by the authors.
> Does the rebuttal answer your questions/concerns? If not, why?
>
> Best,
> AC

---

### Author Rebuttal · Authors · 2024-08-07

# General Response
We sincerely appreciate all reviewers’ time and efforts in reviewing our paper. We are glad to find that reviewers generally recognized our contributions:
-  **Method Development**. Presenting an innovative and intuitive approach, introducing a novel application of VLMs [ovGy, Y6C5]. The perception module is well-designed, with the capacity to acquire skills from observed trajectories being noteworthy [Y6C5].
-  **Comprehensive experiments**. Demonstrating impressive performance across substantial real robots and simulation tasks [Y6C5, xpmL, aTuD, ovGy]. Comprehensive ablation studies are conducted to evaluate various design choices [Y6C5, aTuD].
-  **Writing**. Well-organized and easy to follow [xpmL, Y6C5].

We also thank all reviewers for their insightful and constructive suggestions, which help a lot in further improving our paper. In addition to the pointwise responses below, we summarize supporting experiments and illustrative figures added in the rebuttal according to reviewers’ suggestions.

**New experiments**.

- **Precise manipulation tasks** [Y6C5, xpmL]. Our method demonstrates strong performance on the Functional Manipulation Benchmark (FMB) [1], which encompasses the grasping and reorienting that utilizes the fixture, and high-precision object insertion. Experimental results indicate that our method significantly outperforms CaP. Moreover, our method achieves superior performance to DP, requiring only five human videos compared to DP's 100 robot demonstrations.
- **Robustness against viewpoint variance** [Y6C5]. Experiments are conducted in real-world unseen environments, utilizing four distinct viewpoint configurations. Results demonstrate the significant resilience of VLMimic to viewpoint changes.

**New illustrative figures** in the attached PDF.
- Examples of **VLMimic in complex FMB**, which demands precise manipulation.
- Comparative illustration of **fine-grained motion skill acquisition** between VLMimic and VLM-based planning approaches.
- Examples of **important object reasoning**.
- **Multi-choice QA examples** for grasping constraint reasoning.
- **Failure cases** of VLMimic.
- Illustration of **data generation role** of human-object interaction grounding.
- Visualization of **knowledge bank construction and retrieval**.
- Illustration of **grasping and manipulation phase identification**.


[1] Luo J, Xu C, Liu F, et al. FMB: A functional manipulation benchmark for generalizable robotic learning[J]. arXiv preprint arXiv:2401.08553, 2024.

---

### Author Response · Authors · 2024-08-14

# Summary of rebuttal
We extend our profound gratitude to the Area Chair for their dedicated efforts and to the reviewers for their comprehensive discussions, which have substantially improved the quality of our work. The recommendation for acceptance from all reviewers is particularly gratifying, especially the positive assessment of our method development, comprehensive experiments, and manuscript composition.

This paper is further enriched by additional empirical investigations, encompassing high-precision manipulation tasks and the robustness assessment of our method against variations in viewpoint. Furthermore, new illustrative figures are incorporated to enhance clarity and facilitate understanding for readers.

We reiterate our sincere appreciation to the Area Chair and reviewers for their invested time, diligent effort, and invaluable feedback, which are pivotal in refining our work.

---

### Decision · Program_Chairs · 2024-09-25

**Decision:**

Accept (poster)

**Comment:**

The paper introduces VLMimic, an approach for robotic imitation learning using vision-language models. The method demonstrates strong performance and generalization from few demonstrations, earning recommendation for acceptance from all reviewers.
Reviewers consistently praised the strong performance across diverse tasks, the ability to learn from few demonstrations, comprehensive experiments, and the innovative use of VLMs for fine-grained skill learning. These strengths were evident in both simulation and real-world settings, with VLMimic significantly outperforming baselines.
Initial concerns focused on the complexity of the pipeline, scalability questions, and lack of clarity on technical details. The authors provided thorough responses, clarifying the roles of different components, demonstrating promising scalability, and offering additional implementation details. These responses satisfactorily addressed most reviewer concerns, with two reviewers explicitly improving their scores post-rebuttal.
While some reservations remain about the ad-hoc nature of the design, the overall consensus is that VLMimic represents a valuable contribution to the field. The paper's comprehensive evaluations and strong results outweigh its limitations.
For the final version, I encourage the authors to incorporate their rebuttal explanations, particularly regarding component roles, scalability analysis, and real-world failure cases. Additionally, improving consistency in terminology and addressing the potential for a more generalizable framework would further strengthen the paper.